# Large Scale Markov Decision Processes with Changing Rewards

**Adrian Rivera Cardoso,   He Wang**
School of Industrial and Systems Engineering
Georgia Institute of Technology
adrian.riv@gatech.edu, he.wang@isye.gatech.edu

**Huan Xu**
Alibaba Group
huan.xu@alibaba-inc.com

## Abstract

We consider Markov Decision Processes (MDPs) where the rewards are unknown and may change in an adversarial manner. We provide an algorithm that achieves a regret bound of $O(\sqrt{\tau(\ln|S| + \ln|A|)T}\ln(T))$, where $S$ is the state space, $A$ is the action space, $\tau$ is the mixing time of the MDP, and $T$ is the number of periods. The algorithm's computational complexity is polynomial in $|S|$ and $|A|$. We then consider a setting often encountered in practice, where the state space of the MDP is too large to allow for exact solutions. By approximating the state-action occupancy measures with a linear architecture of dimension $d \ll |S|$, we propose a modified algorithm with a computational complexity polynomial in $d$ and independent of $|S|$. We also prove a regret bound for this modified algorithm, which to the best of our knowledge, is the first $\tilde{O}(\sqrt{T})$ regret bound in the large-scale MDP setting with adversarially changing rewards.

## 1   Introduction

In this paper, we study Markov Decision Processes (hereafter MDPs) with arbitrarily varying rewards. MDP provides a general mathematical framework for modeling sequential decision making under uncertainty [8, 24, 35]. In the standard MDP setting, if the process is in some state $s$, the decision maker takes an action $a$ and receives an expected reward of $r(s, a)$. The process then randomly enters a new state according to some known transition probability. In particular, the standard MDP model assumes that the decision maker has complete knowledge of the reward function $r(s, a)$, which does not change over time.

Over the past two decades, there has been much interest in sequential learning and decision making in an unknown and possibly *adversarial* environment. A wide range of sequential learning problems can be modeled using the framework of Online Convex Optimization (OCO) [45, 20]. In the OCO setting, the decision maker plays a repeated game against an adversary for a given number of rounds. At the beginning of each round indexed by $t$, the decision maker chooses an action $a_t$ from a convex compact set $A$ and the adversary chooses a concave reward function $r_t(\cdot)$, hence a reward of $r_t(a_t)$ is received. After observing the realized reward function, the decision maker chooses its next action $a_{t+1}$ and so on. Since the decision maker does not know the future reward functions, its goal is to achieve a small *regret*; that is, the cumulative reward earned throughout the game should be close to the cumulative reward if the decision maker had been given the benefit of hindsight to choose a fixed

action. We can express the regret for $T$ rounds as

$$\text{Regret}(T) = \max_{a \in A} \sum_{t=1}^{T} r_t(a) - \sum_{t=1}^{T} r_t(a_t).$$

The OCO model has many applications such as universal portfolios [13, 27, 23], online shortest path [38], and online submodular minimization [22]. It is also closely related with areas such as convex optimization [21, 7] and game theory [10]. There are many algorithms that guarantee sublinear regret, e.g., Online Gradient Descent [45], Perturbed Follow the Leader [28], and Regularized Follow the Leader [37, 4]. Compared with the MDP setting, the main difference is that in OCO there is no notion of states, however the payoffs may be chosen by an adversary.

In this work, we study a general problem framework that unites MDP and OCO, which we call the **Online MDP problem**. More specifically, we consider MDPs where the transition probabilities are known but the rewards are sequentially chosen by an adversary.

We list below some canonical motivating examples that can be modeled as Online MDPs.

- Adversarial Multi-Armed Bandits with Constraints [43]: We can generalize the adversarial multi-armed bandits problem with $k$ arms (see Auer et al. [5]) with various constraints such as: restricting the number of times that an arm can be chosen in a given time interval, limiting how we switch between arms, etc. These constraints can be captured easily by defining appropriate states in the Online MDP.

- The Paging Problem [17]: Suppose we are given $n$ pages. A memory can hold at most $k$ ($k < n$) of them. An arbitrary sequence of paging request arrives. A page request is a *hit* if the associated page is in memory, and is a *miss* otherwise. After each request, the decision maker may swap any page in memory by paying some cost. Note that the state of the memory and the swapping decisions can be modeled using MDP. The decision maker's goal is to maximize the number of hits minus the switching costs.

- The $k$-Server Problem [29, 17]: In this classical problem in computer science, there are $k$ servers, represented as points in a metric space. Requests arrive to the metric space, which are also represented as points. As each request arrives, the decision maker can choose to move one of the servers to the requested point. The goal is to minimize the total distance all servers move. If the arrivals of requests are adversarial, this problem can be modeled as an Online MDP problem, where the state represents the position of servers.

Notice that in all of the problems above, the transition probabilities are known, while the adversarial rewards/costs are observed by the decision maker sequentially after each decision epoch. Moreover, in each of these Online MDP problems, the size of the state space may grow exponentially with the number $k$. Some other noteworthy examples are the stochastic inventory control problem [35] and some server queuing problems [14, 3].

## 1.1 Main Results

We propose a new computationally efficient algorithm that achieves near optimal regret for the Online MDP problem. Our algorithm is based on the (dual) linear programming formulation of infinite-horizon average reward MDPs, which uses the occupancy measure of state-action pairs as decision variables. This approach differs from other papers that have studied the Online MDP problem previously, see review in §1.2.

We prove that the algorithm's regret is bounded by $O(\tau + \sqrt{\tau T(\ln |S| + \ln |A|)} \ln(T))$, where $S$ denotes the state space, $A$ denotes the action space, $\tau$ is the mixing time of the MDP, and $T$ is the number of periods. Notice that this regret bound depends *logarithmically* on the size of the state and action space. The algorithm solves a regularized linear program in each period with $poly(|S||A|)$ complexity. The regret bound and the computation complexity compares favorably to the existing methods, which are summarized in §1.2.

We then extend our results to the case where the state space $S$ is extremely large so that $poly(|S||A|)$ computational complexity is impractical. We assume the state-action occupancy measures associated with stationary policies are approximated with a linear architecture of dimension $d \ll |S|$. We design an approximate algorithm combining several innovative techniques for solving large scale MDPs

inspired by [2, 3]. A salient feature of this algorithm is that its computational complexity does not depend on the size of the state-space but instead on the number of features $d$. The algorithm has a regret bound $O(c_{S,A}(\ln |S| + \ln |A|)\sqrt{\tau T} \ln T)$, where $c_{S,A}$ is a problem dependent constant. To the best of our knowledge, this is the first $\tilde{O}(\sqrt{T})$ regret result for large scale Online MDPs.

## 1.2 Related Work

The history of MDP goes back to the seminal work of Bellman [6] and Howard [24] from the 1950's. Some classic algorithms for solving MDP include policy iteration, value iteration, policy gradient, Q-learning and their approximate versions (see [35, 8, 9] for an excellent discussion). In this paper, we will focus on a relatively less used approach, which is based on finding the *occupancy measure* using linear programming, as done recently in [12, 39, 2] to solve MDPs with *static* rewards (see more details in Section 3.1). To deal with the curse of dimensionality, Chen et al. [12] uses bilinear functions to approximate the occupancy measures and Abbasi-Yadkori et al. [2] uses a linear approximation.

The Online MDP problem was first studied a decade ago by [43, 17]. Even-Dar et al. [17] developed no regret algorithms where the bound scales as $O(\tau^2\sqrt{T\ln(|A|)})$, where $\tau$ is the mixing time defined in §2. Their method runs an expert algorithm (e.g. Weighted Majority [31]) on every state where the actions are the experts. However, the authors did not consider the case with large state space in their paper. Yu et al. [43] proposed a more computationally efficient algorithm using a variant of Follow the Perturbed Leader [28], but unfortunately their regret bound becomes $O(|S||A|^2\tau T^{3/4+\epsilon})$. They also considered approximation algorithm for large state space, but did not establish an exact regret bound. The work most closely related to ours is that from Dick et al. [15], where the authors also use a linear programming formulation of MDP similar to ours. However, there seem to be some gaps in the proof of their results.[1] That issue aside, in order to solve large-scale MDPs, their focus is to efficiently solve the quadratic sub-problems that define their iterates efficiently. Instead, we leverage the linear approximation scheme introduced in [2].

Ma et al. [32] also considers Online MDPs with large state space. Under some conditions, they show sublinear regret using a variant of approximate policy iteration, but the regret rate is left unspecified in their paper. Zimin and Neu [44] considered a special class of MDPs called *episodic* MDPs and design algorithms using the occupancy measure LP formulation. Following this line of work, Neu et al. [34] shows that several reinforcement learning algorithms can be viewed as variant of Mirror Descent [25], thus one can establish convergence properties of these algorithms. In [33], the authors considered Online MDPs with bandit feedback and provide an algorithm based on [17]'s with regret of $O(T^{2/3})$. Some other related work can be found in [11, 30, 26].

A more general problem to the Online MDP setting considered here is where the MDP transition probabilities also change in an adversarial manner, which is beyond the scope of this paper. It is believed that this problem is much less tractable computationally; see discussion in [16]. Yu and Mannor [42] studied MDPs with changing transition probabilities, although [33] questions the correctness of their result, as the regret obtained seems to have broken a lower bound. In [19], the authors use a sliding window approach under a particular definition of regret. Abbasi-Yadkori et al. [1] achieved sublinear regret with changing transition probabilities when compared against a restricted policy class.

## 2 Problem Formulation: Online MDP

We consider a general Markov Decision Process (MDP) with known transition probabilities but unknown and adversarially chosen rewards. Let $S$ denote the set of possible states, and $A$ denote the set of actions. (For notational simplicity, we assume the set of actions a player can take is the same for all states, but this assumption can be relaxed easily.) At each period $t \in [T]$, if the system is in state $s_t \in S$, the decision maker chooses an action $a_t \in A$ and collects a reward $r_t(s_t, a_t)$. Here, $r_t : S \times A \to [-1, 1]$ denotes a reward function for period $t$. We assume that the sequence of reward

functions $\{r_t\}_{t=1}^T$ is initially unknown to the decision maker. The function $r_t$ is revealed only after the action $a_t$ has been chosen. We allow the sequence $\{r_t\}_{t=1}^T$ to be chosen by an *adaptive adversary*, meaning $r_t$ can be chosen using the history $\{s_i\}_{i=1}^t$ and $\{a_i\}_{i=1}^{t-1}$. In particular, the adversary does not observe the action $a_t$ when choosing $r_t$. After $a_t$ is chosen, the system then proceeds to state $s_{t+1}$ in the next period with probability $P(s_{t+1}|s_t, a_t)$. We assume the decision maker has complete knowledge of the transition probabilities given by $P(s'|s, a) : S \times A \to S$.

Suppose that the initial state of the MDP follows $s_1 \sim \nu_1$, where $\nu_1$ is a probability distribution over $S$. The objective of the decision maker is to choose a sequence of actions based on the history of states and rewards observed, such that the cumulative reward in $T$ periods is close to that of the optimal offline static policy. Formally, let $\pi$ denote a stationary (possibly randomized) policy: $\pi : S \to \Delta_A$, where $\Delta_A$ is the set of probability distributions over the action set $A$. Let $\Pi$ denote the set of all stationary policies. We aim to find an algorithm that minimizes

$$\text{MDP-Regret}(T) \triangleq \sup_{\pi \in \Pi} R(T, \pi), \text{ with } R(T, \pi) \triangleq \mathbb{E}[\sum_{t=1}^T r_t(s_t^\pi, a_t^\pi)] - \mathbb{E}[\sum_{t=1}^T r_t(s_t, a_t)], \quad (1)$$

where the expectations are taken with respect to random transitions of MDP and (possibly) external randomization of the algorithm.

## 3 Preliminaries

Next, we provide additional notations for the MDP. Let $P_{s,s'}^\pi \triangleq P(s' \mid s, \pi(s))$ be the probability of transitioning from state $s$ to $s'$ given a policy $\pi$. Let $P^\pi$ be an $|S| \times |S|$ matrix with entries $P_{s,s'}^\pi$ ($\forall s, s' \in S$). We use row vector $\nu_t \in \Delta_S$ to denote the probability distribution over states at time $t$. Let $\nu_{t+1}^\pi$ be the distribution over states at time $t + 1$ under policy $\pi$, given by $\nu_{t+1}^\pi = \nu_t P^\pi$. Let $\nu_{st}^\pi$ denote the stationary distribution for policy $\pi$, which satisfies the linear equation $\nu_{st}^\pi = \nu_{st}^\pi P^\pi$. We assume the following condition on the convergence to stationary distribution, which is commonly used in the MDP literature [see 43, 17, 33].

**Assumption 1.** *There exists a real number $\tau \geq 0$ such that for any policy $\pi \in \Pi$ and any pair of distributions $\nu, \nu' \in \Delta_S$, it holds that $\|\nu P^\pi - \nu' P^\pi\|_1 \leq e^{-\frac{1}{\tau}} \|\nu - \nu'\|_1$.*

We refer to $\tau$ in Assumption 1 as the *mixing time*, which measures the convergence speed to the stationary distribution. In particular, the assumption implies that $\nu_{st}^\pi$ is unique for a given policy $\pi$.

We use $\mu(s, a)$ to denote the proportion of time that the MDP visits state-action pair $(s, a)$ in the long run. We call $\mu^\pi \in \mathbb{R}^{|S| \times |A|}$ the *occupancy measure* of policy $\pi$. Let $\rho_t^\pi$ be the long-run average reward under policy $\pi$ when the reward function is fixed to be $r_t$ every period, i.e., $\rho_t^\pi \triangleq \lim_{T \to \infty} \frac{1}{T} \sum_{i=1}^T \mathbb{E}[r_t(s_i^\pi, a_i^\pi)]$. We define $\rho_t \triangleq \rho_t^{\pi_t}$, where $\pi_t$ is the policy selected by the decision maker at time $t$.

### 3.1 Linear Programming Formulation for the Average Reward MDP

Given a reward function $r : S \times A \to [-1, 1]$, suppose one wants to find a policy $\pi$ that maximizes the long-run average reward: $\rho^* = \sup_\pi \lim_{T \to \infty} \frac{1}{T} \sum_{t=1}^T r(s_t^\pi, a_t^\pi)$. Under Assumption 1, the Markov chain induced by any policy is ergodic and the long-run average reward is independent of the starting state (see [8]). It is well known that the optimal policy can be obtained by solving the Bellman equation, which in turn can be written as a linear program (in the dual form):

$$\rho^* = \max_\mu \sum_{s \in S} \sum_{a \in A} \mu(s, a) r(s, a) \quad (2)$$

$$\text{s.t.} \sum_{s \in S} \sum_{a \in A} \mu(s, a) P(s'|s, a) = \sum_{a \in A} \mu(s', a) \quad \forall s' \in S$$

$$\sum_{s \in S} \sum_{a \in A} \mu(s, a) = 1, \quad \mu(s, a) \geq 0 \quad \forall s \in S, \forall a \in A.$$

Let $\mu^*$ be an optimal solution to the LP (2). We can construct an optimal policy of the MDP by defining $\pi^*(s, a) \triangleq \frac{\mu^*(s,a)}{\sum_{a \in A} \mu^*(s,a)}$ for all $s \in S$ such that $\sum_{a \in A} \mu^*(s, a) > 0$; for states where the

denominator is zero, the policy may choose arbitrary actions, since those states will not be visited in the stationary distribution. Let $\nu_{st}^*$ be the stationary distribution over states under this optimal policy.

For simplicity, we will write the first constraint of LP (2) in the matrix form as $\mu^\top (P - B) = 0$, where $B$ is an appropriately chosen matrix with 0-1 entries. We denote the feasible set of the above LP as $\Delta_M \triangleq \{\mu \in \mathbb{R} : \mu \geq 0, \mu^\top 1 = 1, \mu^\top (P - B) = 0\}$. The following definition will be used in the analysis later.

**Definition 1.** *Let $\delta_0 \geq 0$ be the largest real number such that for all $\delta \in [0, \delta_0]$, the set $\Delta_{M,\delta} \triangleq \{\mu \in \mathbb{R}^{|S| \times |A|} : \mu \geq \delta, \mu^\top 1 = 1, \mu^\top (P - B) = 0\}$ is nonempty.*

## 4  A Sublinear Regret Algorithm for Online MDP

In this section, we present an algorithm for the Online MDP problem. The algorithm is very intuitive given the LP formulation (2) for the static problem. As the rewards may change each round, the algorithm simply treats the Online MDP problem as an Online Convex Optimization (OCO) problem with reward functions $\{r_t\}_{t=1}^T$ and decision set $\Delta_M$.

---

**Algorithm 1** (MDP-RFTL)

**input:** parameter $\delta > 0, \eta > 0$, regularization term $R(\mu) = \sum_{s \in S} \sum_{a \in A} \mu(s, a) \ln(\mu(s, a))$

**initialization:** choose any $\mu_1 \in \Delta_{M,\delta} \subset \mathbb{R}^{|S| \times |A|}$

**for** $t = 1, \ldots T$ **do**

    observe current state $s_t$

    **if** $\sum_{a \in A} \mu_t(s_t, a) > 0$ **then**

        choose action $a \in A$ with probability $\frac{\mu_t(s_t, a)}{\sum_a \mu_t(s_t, a)}$.

    **else**

        choose action $a \in A$ with probability $\frac{1}{|A|}$

    **end if**

    observe reward function $r_t \in [-1, 1]^{|S||A|}$

    update $\mu_{t+1} \leftarrow \arg\max_{\mu \in \Delta_{M,\delta}} \sum_{i=1}^t \left[ \langle r_i, \mu \rangle - \frac{1}{\eta} R(\mu) \right]$

**end for**

---

At the beginning of each round $t \in [T]$, the algorithm starts with an occupancy measure $\mu_t$. If the MDP is in state $s_t$, we play action $a \in A$ with probability $\frac{\mu_t(s_t, a)}{\sum_a \mu_t(s_t, a)}$. If the denominator is 0, the algorithm picks any action in $A$ with equal probability. After observing reward function $r_t$ and collecting reward $r_t(s_t, a_t)$, the algorithm changes the occupancy measure to $\mu_{t+1}$.

The new occupancy measure is chosen according to the Regularized Follow the Leader (RFTL) algorithm [37, 4]. RFTL chooses the best occupancy measure for the cumulative reward observed so far, $\sum_{i=1}^t r_i$, plus a regularization term $R(\mu)$. The regularization term forces the algorithm not to drastically change the occupancy measure from round to round. In particular, we choose $R(\mu)$ to be the entropy function. This choice will allow us to get $\ln(|S||A|)$ dependence in the regret bound.

The complete algorithm is shown in Algorithm 1. The main result of this section is the following.

**Theorem 1.** *Suppose $\{r_t\}_{t=1}^T$ is an arbitrary sequence of rewards such that $|r_t(s, a)| \leq 1$ for all $s \in S$ and $a \in A$. For $T \geq \ln^2(1/\delta_0)$, the MDP-RFTL algorithm with parameters $\eta = \sqrt{\frac{T \ln(|S||A|)}{\tau}}$, $\delta = e^{-\sqrt{T}/\sqrt{\tau}}$ guarantees*

$$\text{MDP-Regret}(T) \leq O\left( \tau + 4\sqrt{\tau T (\ln|S| + \ln|A|)} \ln(T) \right).$$

The regret bound in Theorem 1 is near optimal: a lower bound of $\Omega(\sqrt{T \ln|A|})$ exists for the problem of learning with expert advice [18, 20], a special case of Online MDP where the state space is a singleton. We note that the bound only depends *logarithmically* on the size of the state space and action space. The state-of-the-art regret bound for Online MDPs is that of [17], which is $O(\tau + \tau^2 \sqrt{\ln(|A|)T})$. Compared to their result, our bound is better by a factor of $\tau^{3/2}$. However,

our bound has depends on $\sqrt{\ln|S|+\ln|A|}$, whereas the bound in [17] depends on $\sqrt{\ln|A|}$. Both algorithms require $poly(|S||A|)$ computation time, but are based on different ideas: the algorithm of [17] is based on expert algorithms and requires computing $Q$-functions at each time step, whereas our algorithm is based on RFTL. In the next section, we will show how to extend our algorithm to the case with large state space.

### 4.1 Proof Idea for Theorem 1

The key to analyze our algorithm is to decompose the regret with respect to policy $\pi \in \Pi$ as follows

$$R(T,\pi) = \left[\mathbb{E}[\sum_{t=1}^{T} r_t(s_t^\pi, a_t^\pi)] - \sum_{t=1}^{T} \rho_t^\pi\right] + \left[\sum_{t=1}^{T} \rho_t^\pi - \sum_{t=1}^{T} \rho_t\right] + \left[\sum_{t=1}^{T} \rho_t - \mathbb{E}[\sum_{t=1}^{T} r_t(s_t, a_t)]\right]. \quad (3)$$

This decomposition was first used by [17]. We now give some intuition on why $R(T,\pi)$ should be sublinear. By the mixing condition in Assumption 1, the state distribution $\nu_t^\pi$ at time $t$ under a policy $\pi$ differs from the stationary distribution $\nu_{st}^\pi$ by at most $O(\tau)$. This result can be used to bound the first term of (3).

The second term of (3) can be related to the online convex optimization (OCO) problem through the linear programming formulation from §3.1. Notice that $\rho_t^\pi = \sum_{s \in S} \sum_{a \in A} \mu^\pi(s,a) r(s,a) = \langle \mu^\pi, r \rangle$, and $\rho_t = \sum_{s \in S} \sum_{a \in A} \mu_t^\pi(s,a) r(s,a) = \langle \mu^{\pi_t}, r \rangle$. Therefore, we have

$$\sum_{t=1}^{T} \rho_t^\pi - \sum_{t=1}^{T} \rho_t = \sum_{t=1}^{T} \langle \mu^\pi, r_t \rangle - \sum_{t=1}^{T} \langle \mu^{\pi_t}, r_t \rangle, \quad (4)$$

which is exactly the regret quantity commonly studied in the OCO problem. We are thus seeking an algorithm that can bound $\max_{\mu^\pi \in \Delta_M} \sum_{t=1}^{T} \langle \mu^\pi, r_t \rangle - \sum_{t=1}^{T} \langle \mu^{\pi_t}, r_t \rangle$. In order to achieve logarithmic dependence on $|S|$ and $|A|$ in Theorem 1, we apply the RFTL algorithm, regularized by the negative entropy function $R(\mu)$. A technical challenge we faced in the analysis is that $R(\mu)$ is not Lipschitz continuous over the feasible set $\Delta_M$. So we design the algorithm to play in a shrunk set $\Delta_{M,\delta}$ for some $\delta > 0$ (see Definition 1), in which $R(\mu)$ is indeed Lipschitz continuous.

For the last term in (3), note that it is similar to the first term, albeit more complicated: the policy $\pi$ is fixed in the first term, but the policy $\pi_t$ used by the algorithm is varying over time. To solve this challenge, the key idea is to show that the policies do not change too much from round to round, so that the third term grows sublinearly in $T$. To this end, we use the property of the RFTL algorithm with a carefully chosen regularization parameter $\eta > 0$. The complete proof of Theorem 1 can be found in Appendix A.

## 5 Online MDPs with Large State Space

In the previous section, we designed an algorithm for Online MDP with sublinear regret. However, the computational complexity of our algorithm is $O(poly(|S||A|))$ per round. MDPs in practice often have extremely large state space $S$ due to the curse of dimenionality [8], so computing the exact solution becomes impractical. In this section, we propose an approximate algorithm that can handle large state space.

### 5.1 Approximating Occupancy Measures and Regret Definition

We consider an approximation scheme introduced in [3] for standard MDPs. The idea is to use $d$ feature vectors (with $d \ll |S||A|$) to approximate occupancy measures $\mu \in \mathbb{R}^{|S| \times |A|}$. Specifically, we approximate $\mu \approx \Phi\theta$ where $\Phi$ is a given matrix of dimension $|S||A| \times d$, and $\theta \in \Theta \triangleq \{\theta \in \mathbb{R}_+^d : \|\theta\|_\infty \leq W\}$ for some positive constant $W$. As we will restrict the occupancy measures chosen by our algorithm to satisfy $\mu = \Phi\theta$, the definition of MDP-regret (1) is too strong as it compares against all stationary policies. Instead, we restrict the benchmark to be the set of policies $\Pi^\Phi$ that can be represented by matrix $\Phi$, where

$$\Pi^\Phi \triangleq \{\pi \in \Pi : \text{ there exists } \mu^\pi \in \Delta_M \text{ such that } \mu^\pi = \Phi\theta \text{ for some } \theta \in \Theta\}.$$

Our goal will now be to achieve sublinear $\Phi$-MDP-regret defined as

$$\Phi\text{-MDP-Regret}(T) \triangleq \max_{\pi \in \Pi^\Phi} \mathbb{E}[\sum_{t=1}^T r_t(s_t^\pi, a_t^\pi)] - \mathbb{E}[\sum_{t=1}^T r_t(s_t, a_t)], \tag{5}$$

where the expectation is taken with respect to random state transitions of the MDP and randomization used in the algorithm. Additionally, we want to make the computational complexity *independent* of $|S|$ and $|A|$.

**Choice of Matrix $\Phi$ and Computation Efficiency.** The columns of matrix $\Phi \in \mathbb{R}^{|S||A| \times d}$ represent probability distributions over state-action pairs. The choice of $\Phi$ is problem-dependent, and a detailed discussion is beyond the scope of this paper. Abbasi-Yadkori et al. [3] shows that for many applications such as the game of Tetris and queuing networks, $\Phi$ can be naturally chosen as a sparse matrix, which allows constant time access to entries of $\Phi$ and efficient dot product operations. We will assume such constant time access throughout our analysis. We refer readers to [3] for further details.

## 5.2 The Approximate Algorithm

The algorithm we propose is built on MDP-RFTL, but is significantly modified in several aspects. We start with key ideas on how and why we need to modify the previous algorithm, and then formally present the new algorithm. To aid our analysis, we make the following definition.

**Definition 2.** *Let $\tilde{\delta}_0 \geq 0$ be the largest real number such that for all $\delta \in [0, \tilde{\delta}_0]$ the set $\Delta_{M,\delta}^\Phi \triangleq \{\mu \in \mathbb{R}^{|S||A|} : \text{there exists } \theta \in \Theta \text{ such that } \mu = \Phi\theta, \mu \geq \delta, \mu^\top 1 = 1, \mu^\top(P - B) = 0\}$ is nonempty. We also write $\Delta_M^\Phi \triangleq \Delta_{M,0}^\Phi$.*

As a first attempt, one could replace the shrunk set of occupancy measures $\Delta_{M,\delta}$ in Algorithm 1 with $\Delta_{M,\delta}^\Phi$ defined above. We then use occupancy measures $\mu^{\Phi\theta_{t+1}^*} \triangleq \Phi\theta_{t+1}^*$ given by the RFTL algorithm, i.e., $\theta_{t+1}^* \leftarrow \arg\max_{\theta \in \Delta_{M,\delta}^\Phi} \sum_{i=1}^t [\langle r_i, \mu \rangle - (1/\eta)R(\mu)]$. The same proof of Theorem 1 would apply and guarantee a sublinear $\Phi$-MDP-Regret. Unfortunately, replacing $\Delta_{M,\delta}$ with $\Delta_{M,\delta}^\Phi$ does not reduce the time complexity of computing the iterates $\{\mu^{\Phi\theta_t^*}\}_{t=1}^T$, which is still $poly(|S||A|)$.

To tackle this challenge, we will not apply the RFTL algorithm exactly, but will instead obtain an approximate solution in $poly(d)$ time. We relax the constraints $\mu \geq \delta$ and $\mu^\top(P - B) = 0$ that define the set $\Delta_{M,\delta}^\Phi$, and add the following penalty term to the objective function:

$$V(\theta) \triangleq -H_t \|(\Phi\theta)^\top(P - B)\|_1 - H_t \|\min\{\delta, \Phi\theta\}\|_1. \tag{6}$$

Here, $\{H_t\}_{t=1}^T$ is a sequence of tuning parameters that will be specified in Theorem 2. Let $\Theta^\Phi \triangleq \{\theta \in \Theta, \mathbf{1}^\top(\Phi\theta) = 1\}$. Thus, the original RFTL step in Algorithm 1 now becomes

$$\max_{\theta \in \Theta^\Phi} \sum_{i=1}^t c^{t,\eta}(\theta), \quad \text{where } c^{t,\eta}(\theta) \triangleq \sum_{i=1}^t \left[ \langle r_i, \Phi\theta \rangle - \frac{1}{\eta} R^\delta(\Phi\theta) \right] + V(\theta). \tag{7}$$

In the above function, we use a modified entropy function $R^\delta(\cdot)$ as the regularization term, because the standard entropy function has an infinite gradient at the origin. More specifically, let $R_{(s,a)}(\mu) \triangleq \mu(s,a)\ln(\mu(s,a))$ be the entropy function. We define $R^\delta(\mu) = \sum_{(s,a)} R_{(s,a)}^\delta(\mu(s,a))$, where

$$R_{(s,a)}^\delta \triangleq \begin{cases} R_{(s,a)}(\mu) & \text{if } \mu(s,a) \geq \delta \\ R_{(s,a)}(\delta) + \frac{d}{d\mu(s,a)} R_{(s,a)}(\delta)(\mu(s,a) - \delta) & \text{otherwise.} \end{cases} \tag{8}$$

Since computing an exact gradient for function $c^{t,\eta}(\cdot)$ would take $O(|S||A|)$ time, we solve problem (7) by stochastic gradient ascent. The following lemma shows how to efficiently generate stochastic subgradients for function $c^{t,\eta}$ via sampling.

**Lemma 1.** *Let $q_1$ be any probability distribution over state-action pairs, and $q_2$ be any probability distribution over all states. Sample a pair $(s', a') \sim q_1$ and $s'' \sim q_2$. The quantity*

$$g_{s',a',s''}(\theta) = \Phi^\top r_t + \frac{H_t}{q_1(s',a')}\Phi_{(s',a'),:}\mathbb{I}\{\Phi_{(s',a'),:}\theta \leq \delta\}$$

$$- \frac{H_t}{q_2(s'')}[(P-B)^\top\Phi]_{s'',:}sign([(P-B)^\top\Phi]_{s'',:}\theta) - \frac{t}{\eta q_1(s',a')}\nabla_\theta R^\delta_{(s',a')}(\Phi\theta)$$

*satisfies $\mathbb{E}_{(s',a')\sim q_1, s''\sim q_2}[g_{s',a',s''}(\theta)|\theta] = \nabla_\theta c^{\eta,t}(\theta)$ for any $\theta \in \Theta$. Morever, we have $\|g(\theta)\|_2 \leq t\sqrt{d} + H_t(C_1 + C_2) + \frac{t}{\eta}(1 + \ln(Wd) + |\ln(\delta)|)C_1$, w.p.1, where*

$$C_1 = \max_{(s,a)\in S\times A}\frac{\|\Phi_{(s,a),:}\|_2}{q_1(s,a)}, \quad C_2 = \max_{s\in S}\frac{\|(P-B)_{:,s}^\top\Phi\|_2}{q_2(s)}. \tag{9}$$

Putting everything together, we present the complete approximate algorithm for large state online MDPs in Algorithm 2. The algorithm uses Projected Stochastic Gradient Ascent (Algorithm 3) as a subroutine, which uses the sampling method in Lemma 1 to generate stochastic sub-gradients.

---

**Algorithm 2** (LARGE-MDP-RFTL)

> **input:** matrix $\Phi$, parameters: $\eta, \delta > 0$, convex function $R^\delta(\mu)$, SGA step-size schedule $\{w_t\}_{t=0}^T$, penalty term parameters $\{H_t\}_{t=1}^T$
> **initialize:** $\tilde{\theta}_1 \leftarrow \text{PSGA}(-R^\delta(\Phi\theta) + V(\theta), \Theta^\Phi, w_0, K_0)$
> **for** $t = 1, ..., T$ **do**
> > observe current state $s_t$; play action $a$ with distribution $\frac{[\Phi\tilde{\theta}_t]_+(s_t,a)}{\sum_{a\in A}[\Phi\tilde{\theta}_t]_+(s_t,a)}$
> > observe $r_t \in [-1,1]^{|S||A|}$
> > $\tilde{\theta}_{t+1} \leftarrow \text{PSGA}(\sum_{i=1}^t[\langle r_i, \Phi\theta\rangle - \frac{1}{\eta}R^\delta(\Phi\theta)] + V(\theta), \Theta^\Phi, w_t, K_t)$
> **end for**

---

**Algorithm 3** Projected Stochastic Gradient Ascent: $\text{PSGA}(f, X, w, K)$

> **input:** concave objective function $f$, feasible set $X$, stepsize $w$, $x_1 \in X$
> **for** $k = 1, ...K$ **do**
> > compute a stochastic subgradient $g_k$ such that $\mathbb{E}[g_k] = \nabla f(x_k)$ using Lemma 1
> > set $x_{k+1} \leftarrow P_X(x_k + wg(x_k))$
> **end for**
> **output:** $\frac{1}{K}\sum_{k=1}^K x_k$

---

### 5.3 Analysis of the Approximate Algorithm

We establish a regret bound for the LARGE-MDP-RFTL algorithm as follows.

**Theorem 2.** *Suppose $\{r_t\}_{t=1}^T$ is an arbitrary sequence of rewards such that $|r_t(s,a)| \leq 1$ for all $s \in S$ and $a \in A$. For $T \geq \ln^2(\frac{1}{\delta_0})$, LARGE-MDP-RFTL with parameters $\eta = \sqrt{\frac{T}{\tau}}, \delta = e^{-\sqrt{T}}$, $K(t) = [W^{3/2}t^2d^{3/2}\tau^4(C_1+C_2)T^{3/2}\ln(WTd)]^2$, $w_t = \frac{\sqrt{d}W}{\sqrt{K(t)}(t\sqrt{d}+H_t(C_1+C_2)+\frac{t}{\eta}C_1)}$ guarantees that*

$$\Phi\text{-MDP-Regret}(T) \leq O(c_{S,A}\ln(|S||A|)\sqrt{\tau T}\ln(T)).$$

*Here $c_{S,A}$ is a problem dependent constant. The constants $C_1, C_2$ are defined in Lemma 1.*

A salient feature of the LARGE-MDP-RFTL algorithm is that its computational complexity in each period is independent of the size of state space $|S|$ or the size of action space $|A|$, and thus is amenable to large scale MDPs. In particular, in Theorem 2, the number of SGA iterations, $K(t)$, is $O(d)$ and independent of $|S|$ and $|A|$.

Compared to Theorem 1, we achieve a regret with similar dependence on the number of periods $T$ and the mixing time $\tau$. The regret bound also depends on $\ln(|S|)$ and $\ln(|A|)$, with an additional

constant term $c_{S,A}$. The constant comes from a projection problem (see details in Appendix B) and may grow with $|S|$ and $|A|$ in general. But for some MDP problems, $c_{S,A}$ can be bounded by an absolute constant: an example is the well-known (Markovian) multi-armed bandit problem [41]. For a more detailed discussion of the constant $c_{S,A}$, we refer readers to Appendix C.

**Proof Idea for Theorem 2.** Consider the MDP-RFTL iterates ,$\{\theta_t^*\}_{t=1}^T$, and the occupancy measures $\{\mu^{\Phi\theta_t^*}\}_{t=1}^T$ induced by following policies $\{\Phi\theta_t^*\}_{t=1}^T$. Since $\theta_t^* \in \Delta_{M,\delta}^\Phi$ it holds that $\mu^{\Phi\theta_t^*} = \Phi\theta_t^*$ for all $t$. Thus, following the proof of Theorem 1, we can obtain the same $\Phi$-MDP-Regret bound in Theorem 1 if we follow policies $\{\Phi\theta_t^*\}_{t=1}^T$. However, computing $\theta_t^*$ takes $O(poly(|S||A|))$ time.

The crux the proof of Theorem 2 is to show that the $\{\Phi\tilde{\theta}_t\}_{t=1}^T$ iterates in Algorithm 2 induce occupancy measures $\{\mu^{\Phi\tilde{\theta}_t}\}_{t=1}^T$ that are close to $\{\mu^{\Phi\theta_t^*}\}_{t=1}^T$. Since the algorithm has relaxed constraints of $\Delta_{M,\delta}^\Phi$, in general we have $\Phi\tilde{\theta}_t \notin \Delta_{M,\delta}^\Phi$ and thus $\mu^{\Phi\tilde{\theta}_t} \neq \Phi\tilde{\theta}_t$. So we need to show that the distance between $\mu^{\Phi\theta_{t+1}^*}$, and $\mu^{\Phi\tilde{\theta}_{t+1}}$ is small. Using triangle inequality we have

$$\|\mu^{\Phi\theta_t^*} - \mu^{\Phi\tilde{\theta}_t}\|_1 \leq \|\mu^{\Phi\theta_t^*} - P_{\Delta_{M,\delta}^\Phi}(\Phi\tilde{\theta}_t)\|_1 + \|P_{\Delta_{M,\delta}^\Phi}(\Phi\tilde{\theta}_t) - \Phi\tilde{\theta}_t\|_1 + \|\Phi\tilde{\theta}_t - \mu^{\Phi\tilde{\theta}_t}\|_1,$$

where $P_{\Delta_{M,\delta}^\Phi}(\cdot)$ denotes the Euclidean projection onto $\Delta_{M,\delta}^\Phi$. We then proceed to bound each term individually. We defer the details to Appendix B as bounding each term requires lengthy proofs.

# 6    Conclusion

We consider Markov Decision Processes (MDPs) where the transition probabilities are known but the rewards are unknown and may change in an adversarial manner. We provide a simple online algorithm, which applies Regularized Follow the Leader (RFTL) to the linear programming formulation of the average reward MDP. The algorithm achieves a regret bound of $O(\sqrt{\tau(\ln|S| + \ln|A|)T}\ln(T))$, where $S$ is the state space, $A$ is the action space, $\tau$ is the mixing time of the MDP, and $T$ is the number of periods. The algorithm's computational complexity is polynomial in $|S|$ and $|A|$ per period.

We then consider a setting often encountered in practice, where the state space of the MDP is too large to allow for exact solutions. We approximate the state-action occupancy measures with a linear architecture of dimension $d \ll |S||A|$. We then propose an approximate algorithm that relaxes the constraints in the RFTL algorithm, and then solve the relaxed problem using stochastic gradient descent method. The computational complexity of this approximate algorithm is independent of the size of state space $|S|$ and the size of action space $|A|$. We prove a regret bound of $O(c_{S,A}\ln(|S||A|)\sqrt{\tau T}\ln(T))$ compared to the best static policy approximated by the linear architecture, where $c_{S,A}$ is a problem dependent constant. To the best of our knowledge, this is the first $\tilde{O}(\sqrt{T})$ regret bound for large scale MDPs with changing rewards.

## Footnotes

[1] In particular, we believe the proof of Lemma 1 in [15] is incorrect. Equation (8) in their paper states that the regret relative to a policy is equal to the sum of a sequence of vector products; however, the dimensions of vectors involved in these dot products are incompatible. By their definition, the variable $\nu_t$ is a vector of dimension $|S|$, which is being multiplied with a loss vector with dimension $|S||A|$.

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
