[Supplementary Material · online_mdps-12-26.pdf]

# A    Regret Analysis of MDP-RFTL: Proof of Theorem 1

To bound the regret incurred by MDP-RFTL, we bound each term in Eq (3). We start with the first term. We use the following lemma, which was first stated in [17] and was also used by [33].

**Lemma 2.** *For any $T \geq 1$ and any policy $\pi$ it holds that*

$$\mathbb{E}[\sum_{t=1}^{T} r_t(s_t^\pi, a_t^\pi)] - \sum_{t=1}^{T} \rho_t^\pi \leq 2\tau + 2.$$

*Proof of Lemma 2 .* Recall that $|r_t(s, a)| \leq 1$, so we have $|\sum_{a \in A} \pi(s, a) r_t(s, a)| \leq 1$ by Cauchy-Schwarz inequality, since $\pi(s, \cdot)$ defines a probability distribution over actions. Also, recall that $\nu_t^\pi$ is the stationary distribution over states by following policy $\pi$ and $\nu_{t+1}^\pi = \nu_t^\pi P^\pi$ for all $t \in [T]$. We have

$$\mathbb{E}[\sum_{t=1}^{T} r_t(s_t^\pi, a_t^\pi)] - \sum_{t=1}^{T} \rho_t^\pi = \sum_{t=1}^{T} \sum_{s \in S} (\nu_t^\pi(s) - \nu_{st}^\pi(s)) \sum_{a \in A} \pi(s, a) r_t(s, a)$$

$$\leq \sum_{t=1}^{T} \sum_{s \in S} \nu_t^\pi(s) - \nu_{st}^\pi(s)$$

$$\leq \sum_{t=1}^{T} \|\nu_t^\pi(s) - \nu_{st}^\pi(s)\|_1.$$

Now, notice that

$$\|\nu_t^\pi(s) - \nu_{st}^\pi(s)\|_1 = \|\nu_{t-1}^\pi P^\pi - \nu_{st} P^\pi\|_1$$

$$\leq e^{-\frac{1}{\tau}} \|\nu_{t-1}^\pi - \nu_{st}\|_1 \quad \text{by Assumption 1}$$

$$\leq e^{-\frac{t}{\tau}} \|\nu_1^\pi - \nu_{st}^\pi\|_1 \quad \text{by repeating the argument } t-1 \text{ more times}$$

$$\leq 2e^{-\frac{t}{\tau}}.$$

Finally, we have that

$$\sum_{t=1}^{T} \|\nu_t^\pi(s) - \nu_{st}^\pi(s)\|_1 \leq 2 \sum_{t=1}^{T} e^{-\frac{t}{\tau}}$$

$$\leq 2(1 + \int_0^\infty e^{-\frac{t}{\tau}}) dt$$

$$= 2\tau + 2,$$

which concludes the proof. $\qquad \square$

We now bound the third term in (3). We use the following lemma, which bounds the difference of two stationary distributions by the difference of the corresponding occupancy measures.

**Lemma 3.** *Let $\nu_{st}^1$ and $\nu_{st}^2$ be two arbitrary stationary distributions over $S$. Let $\mu^1$ and $\mu^2$ be the corresponding occupancy mesures. It holds that*

$$\|\nu_{st}^1 - \nu_{st}^2\|_1 \leq \|\mu^1 - \mu^2\|_1.$$

*Proof of Lemma 3.*

$$\|\nu_{st}^1 - \nu_{st}^2\|_1 = \sum_{s \in S} |\nu_{st}^1(s) - \nu_{st}^2(s)|$$

$$= \sum_{s \in S} |\sum_{a \in A} \mu^1(s, a) - \mu^2(s, a)|$$

$$\leq \sum_{s \in S} \sum_{a \in A} |\mu^1(s, a) - \mu^2(s, a)|$$

$$= \|\mu^1 - \mu^2\|_1.$$

$\qquad \square$

We are ready to bound the third term in (3).

**Lemma 4.** *Let $\{s_t, a_t\}_{t=1}^{T}$ be the random sequence of state-action pairs generated by the policies induced by occupancy measures $\{\mu^{\pi_t}\}_{t=1}^{T}$. It holds that*

$$\sum_{t=1}^{T} \rho_t - \mathbb{E}\left[\sum_{t=1}^{T} r_t(s_t, a_t)\right] \leq \sum_{t=1}^{T} 2e^{-\frac{t-1}{\tau}} + \sum_{t=1}^{T}\sum_{\theta=0}^{t-1} e^{-\frac{\theta}{\tau}} \|\mu^{\pi_{t-\theta}} - \mu^{\pi_{t-(\theta+1)}}\|_1.$$

*Proof of Lemma 4.* By the definition of $\rho_t$, we have

$$\sum_{t=1}^{T} \rho_t - \mathbb{E}\left[\sum_{t=1}^{T} r_t(s_t, a_t)\right] = \sum_{t=1}^{T}\sum_{s \in S}(\nu_{st}^{\pi_t}(s) - \nu^t(s))\sum_{a \in A}\pi^t(s,a)r_t(s,a)$$

$$\leq \sum_{t=1}^{T} \|\nu_{st}^{\pi_t} - \nu^t\|_1.$$

Now, recall that $\nu_t = \nu_1 P^{\pi_1} P^{\pi_2}...P^{\pi_{t-1}}$. We now bound $\|\nu_{st}^{\pi_t} - \nu^t\|_1$ for all $t \in [T]$ as follows:

$$\|\nu^t - \nu_{st}^{\pi_t}\|_1 \leq \|\nu^t - \nu_{st}^{\pi_{t-1}}\|_1 + \|\nu_{st}^{\pi_{t-1}} - \nu_{st}^{\pi_t}\|_1$$

$$\leq \|\nu^t - \nu_{st}^{\pi_{t-1}}\|_1 + \|\mu^{\pi_{t-1}} - \mu^{\pi_t}\|_1 \quad \text{by Lemma 3}$$

$$= \|\nu^{t-1}P^{\pi_{t-1}} - \nu_{st}^{\pi_{t-1}}P^{\pi_{t-1}}\|_1 + \|\mu^{\pi_{t-1}} - \mu^{\pi_t}\|_1$$

$$\leq e^{-\frac{1}{\tau}}\|\nu^{t-1} - \nu_{st}^{\pi_{t-1}}\|_1 + \|\mu^{\pi_{t-1}} - \mu^{\pi_t}\|_1 \quad \text{by Assumption 1}$$

$$\leq e^{-\frac{1}{\tau}}(e^{-\frac{1}{\tau}}\|\nu^{t-2} - \nu_{st}^{\pi_{t-2}}\|_1 + \|\mu^{\pi_{t-2}} - \mu^{\pi_{t-1}}\|_1) + \|\mu^{\pi_{t-1}} - \mu^{\pi_t}\|_1$$

$$\leq e^{-\frac{t-1}{\tau}}\|\nu^1 - \nu_{st}^{\pi_1}\|_1 + \sum_{\theta=0}^{t-1} e^{-\frac{\theta}{\tau}}\|\mu^{\pi_{t-\theta}} - \mu^{\pi_{t-(\theta+1)}}\|_1,$$

which yields the desired claim. $\qquad\qquad\square$

Combining Lemma 2, Lemma 4 and Eq (3), we have arrived at the following bound on the regret:

$$R(T,\pi) \leq (2\tau+2) + \left[\sum_{t=1}^{T}\langle\mu^{\pi}, r_t\rangle - \sum_{t=1}^{T}\langle\mu^{\pi_t}, r_t\rangle\right] + \left[\sum_{t=1}^{T} 2e^{-\frac{t-1}{\tau}} + \sum_{t=1}^{T}\sum_{\theta=0}^{t-1} e^{-\frac{\theta}{\tau}}\|\mu^{\pi_{t-\theta}} - \mu^{\pi_{t-(\theta+1)}}\|_1\right].$$

To complete the proof, we want to bound the second and the third terms. For the second term $\max_{\mu \in \Delta_M}\sum_{t=1}^{T}\langle\mu^{\pi}, r_t\rangle - \sum_{t=1}^{T}\langle\mu^{\pi_t}, r_t\rangle$, since the reward functions are linear in $\mu$ and the set $\Delta_M$ is convex, any algorithm for Online Linear Optimization, e.g., Online Gradient Ascent [45], ensures a regret bound that is sublinear $T$. However, this would yield an MDP-regret rate that depends linearly on $|S| \times |A|$.

Instead, by noticing that the feasible set of the LP, $\Delta_M$, is a subset of the probability simplex $\Delta^{|S||A|}$, we use RFTL and regularize using the negative entropy function. This will give us a rate that scales as $\ln(|S||A|)$, which is much more desirable than $O(|S||A|)$. Notice that the algorithm does not work with the set $\Delta_M$ directly but with $\Delta_{M,\delta}$ instead, this is because the negative entropy is not Lipschitz over $\Delta_M$. Working over $\Delta_{M,\delta}$ is the key to being able to bound the third term in the regret decomposition. Formally, we have the following result.

**Lemma 5.** *Let $\{\mu_t\}_{t=1}^{T}$ be the iterates of MDP-RFTL, then it holds that*

$$\max_{\mu \in \Delta_{M,\delta}}\sum_{t=1}^{T}\langle r_t, \mu\rangle \leq \sum_{t=1}^{T}\langle r_t, \mu^{\pi_{t+1}}\rangle + \frac{T}{\eta}\max_{\mu_1,\mu_2 \in \Delta_{M,\delta}}[R(\mu_1) - R(\mu_2)].$$

*Proof of Lemma 5.* Define $f_t \triangleq \langle\mu, r_t\rangle$ and $f_t^R \triangleq f_t(\mu) - \frac{1}{\eta}R(\mu)$ for all $t = 1,..,T$. We first prove by induction that

$$\max_{\mu \in \Delta_{M,\delta}}\sum_{t=1}^{T} f_t^R(\mu) \leq \sum_{t=1}^{T} f_t^R(\mu^{\pi_{t+1}}).$$

The base case $T = 1$ is trivial by the definition of $\mu^{\pi_2}$. Suppose the claim holds for $T - 1$. For all $\mu \in \Delta_{M,\delta}$, we have that

$$
\sum_{t=1}^{T} f_t^R(\mu) \leq \sum_{t=1}^{T} f_t^R(\mu^{\pi_{T+1}})
$$

$$
\leq \max_{\mu \in \Delta_{M,\delta}} \sum_{t=1}^{T-1} f_t^R(\mu) + f_T^R(\mu^{\pi_{T+1}})
$$

$$
\leq \sum_{t=1}^{T-1} f_t^R(\mu^{\pi_{t+1}}) + f_T^R(\mu^{\pi_{T+1}}) \quad \text{by induction hyposthesis}
$$

$$
= \sum_{t=1}^{T} f_t^R(\mu^{\pi_{t+1}}).
$$

The lemma follows by plugging back in the definition of $f_t^R$ and rearranging terms. $\qquad\square$

**Lemma 6.** *Let $\{\mu_t\}_{t=1}^{T}$ be the iterates of MDP-RFTL, it holds that*

$$
\|\mu^{\pi_t} - \mu^{\pi_{t+1}}\|_1 \leq \frac{2\eta}{t}\left(1 + \frac{1}{\eta}G_R\right).
$$

*Proof of Lemma 6.* Let $J(\mu) = \sum_{\theta=1}^{t}\left[\langle\mu, r_t\rangle - \frac{1}{\eta}R(\mu)\right]$. Since $R$ is the negative entropy we know it is 1- strongly convex with respect to norm $\|\cdot\|_1$, thus $J$ is $\frac{t}{\eta}$-strongly concave. By strong concavity we have

$$
\frac{t}{2\eta}\|\mu^{\pi_{t+1}} - \mu^{\pi_t}\|_1^2 \leq J(\mu^{\pi_{t+1}}) - J(\mu^{\pi_t}) + \langle\nabla_\mu J(\mu^{\pi_{t+1}}), \mu^{\pi_t} - \mu^{\pi_{t+1}}\rangle.
$$

Since $\mu^{\pi_{t+1}}$ is the optimizer of $J$ the optimality condition states that $\langle\nabla_\mu J(\mu^{\pi_{t+1}}), \mu^{\pi_t} - \mu^{\pi_{t+1}}\rangle \leq 0$. Plugging back in the definition of $J$ we have that

$$
\frac{t}{2\eta}\|\mu^{\pi_{t+1}} - \mu^{\pi_t}\|_1^2
$$

$$
\leq \sum_{\theta=1}^{t}\left[\langle r_\theta, \mu^{\pi_{t+1}}\rangle - \frac{1}{\eta}R(\mu^{\pi_{t+1}})\right] - \sum_{\theta=1}^{t}\left[\langle r_\theta, \mu^{\pi_t}\rangle - \frac{1}{\eta}R(\mu^{\pi_t})\right]
$$

$$
= \sum_{\theta=1}^{t-1}\left[\langle r_\theta, \mu^{\pi_{t+1}}\rangle - \frac{1}{\eta}R(\mu^{\pi_{t+1}})\right] - \sum_{\theta=1}^{t-1}\left[\langle r_\theta, \mu^{\pi_t}\rangle - \frac{1}{\eta}R(\mu^{\pi_t})\right]
$$

$$
+ \langle r_t, \mu^{\pi_{t+1}}\rangle - \frac{1}{\eta}R(\mu^{\pi_{t+1}}) - \langle r_\theta, \mu^{\pi_t}\rangle + \frac{1}{\eta}R(\mu^{\pi_t})
$$

$$
\leq \langle r_t, \mu^{\pi_{t+1}}\rangle - \frac{1}{\eta}R(\mu^{\pi_{t+1}}) - \langle r_\theta, \mu^{\pi_t}\rangle + \frac{1}{\eta}R(\mu^{\pi_t}) \quad \text{by definition of } \mu^{\pi_t}
$$

$$
\leq \|r_t\|_\infty\|\mu^{\pi_t} - \mu^{\pi_{t+1}}\|_1 + \frac{1}{\eta}R(\mu^{\pi_t}) - \frac{1}{\eta}R(\mu^{\pi_{t+1}}) \quad \text{by Cauchy-Schwarz inequality}
$$

$$
\leq \|\mu^{\pi_t} - \mu^{\pi_{t+1}}\|_1 + \frac{G_R}{\eta}\|\mu^{\pi_t} - \mu^{\pi_{t+1}}\|_1 \quad \text{Since } R \text{ is } G_R\text{- Lipschitz.}
$$

By rearranging terms, we get

$$
\|\mu^{\pi_t} - \mu^{\pi_{t+1}}\|_1 \leq \frac{2\eta}{t}\left(1 + \frac{1}{\eta}G_R\right).
$$

$\qquad\square$

Notice that by Lemma 6 we will need the regularizer $R$ to be Lipschitz continuous with respect to norm $\|\cdot\|_1$. Unfortunately, the negative entropy function is not Lipschitz continuous over $\Delta_M$, so we will force the algorithm to play in a shrunk set $\Delta_{M,\delta}$.

**Lemma 7.** *Let $\Delta_\delta \triangleq \{x \in \mathbb{R}^d : \|x\|_1 = 1, x_i \geq \delta \ \forall i = 1, ..., d\}$. The function $R(x) \triangleq \sum_{i=1}^d x_i \ln(x_i)$ is $G_R$-Lipschitz continuous with respect to $\|\cdot\|_1$ over $\Delta_\delta$ with $G_R = \max\{|\ln(\delta)|, 1\}$.*

*Proof of Lemma 7.* We want to find $G_R > 0$ such that $\|\nabla R(x)\|_\infty \leq G_R$ for all $x \in \Delta_\delta$. Notice that $[\nabla R(x)]_i = 1 + \ln(x_i)$ for $i = 1, ...d$. Moreover, since for every $i = 1, ..., d$ we have $\delta \leq x_i \leq 1$ the following sequence of inequalities hold: $\ln(\delta) \leq 1 + \ln(\delta) \leq 1 + \ln(x_i) \leq 1$. It follows that $G_R = \max\{|\ln(\delta)|, 1\}$. $\qquad\square$

The next lemma quantifies the loss in the regret due to playing in the shrunk set.

**Lemma 8.** *It holds that*

$$\max_{\mu \in \Delta_M} \sum_{t=1}^T \langle r_t, \mu \rangle \leq \max_{\mu \in \Delta_{M,\delta}} \sum_{t=1}^T \langle r_t, \mu \rangle + 2\delta T \left(|S||A| - 1\right).$$

*Proof of Lemma 8.* Given $z^* \in \Delta \subset \mathbb{R}^d$, define $z_p^* \triangleq \arg\min_{z \in \Delta_\delta} \|z - z^*\|_1$, with $\delta \leq \frac{1}{d}$. It holds that $\|z_p^* - z^*\|_1 \leq 2\delta(d-1)$. To see why the previous is true, choose $z^* = [1; 0; 0; ...; 0; 0]$. It is easily verified that $z_p^* = [1 - \delta(d-1); \delta; \delta; ...; \delta, \delta]$ and $\|z^* - z_p^*\|_1 = 2\delta(d-1)$. Because of the previous argument, if $\mu^* \in \arg\max_{\mu \in \Delta_M} \sum_{t=1}^T \langle r_t, \mu \rangle$ and $\mu_p^*$ is its $\|\cdot\|_1$ projection onto the set $\Delta_{M,\delta}$ then $\|\mu^* - \mu_p^*\|_1 \leq 2\delta(|S||A| - 1)$. The claim then follows since each function $\langle r_t, \mu \rangle$ is 1-Lipschitz continuous with respect to $\|\cdot\|_1$.

$\qquad\square$

Given that we know the iterates of MDP-RFTL are close by Lemma 6, we can bound the last term in our regret bound

**Lemma 9.** *It holds that*

$$\sum_{t=1}^T 2e^{-\frac{t-1}{\tau}} + \sum_{t=1}^T \sum_{\theta=0}^{t-1} e^{-\frac{\theta}{\tau}} \|\mu^{\pi_{t-\theta}} - \mu^{\pi_{t-(\theta+1)}}\|_1 \leq 2(1+\tau) + 2\eta\left(1 + \frac{1}{\eta}G_R\right)(1 + \ln(T))(1+\tau).$$

*Proof of Lemma 9.* We first bound the first term

$$\sum_{t=1}^T 2e^{-\frac{t-1}{\tau}} \leq 2(1 + \int_1^\infty e^{-\frac{x-1}{\tau}} \, dx) \leq 2(1 + \tau).$$

We now bound the second term, let $\alpha = 2\eta\left(1 + \frac{1}{\eta}G_R\right)$. We have that

$$
\begin{aligned}
\sum_{t=1}^T \sum_{\theta=0}^{t-1} e^{-\frac{\theta}{\tau}} \|\mu^{\pi_{t-\theta}} - \mu^{\pi_{t-(\theta+1)}}\|_1 &= \alpha \sum_{t=1}^T \sum_{\theta=0}^{t-1} e^{-\frac{\theta}{\tau}} \frac{1}{t-\theta} \\
&= \alpha \left[ e^{-\frac{0}{\tau}} \sum_{t=1}^T 1/t + e^{-\frac{1}{\tau}} \sum_{t=1}^{T-1} 1/t + e^{-\frac{2}{\tau}} \sum_{t=1}^{T-2} 1/t + ... \right] \\
&\leq \alpha \left[ e^{-\frac{0}{\tau}} \sum_{t=1}^T 1/t + e^{-\frac{1}{\tau}} \sum_{t=1}^T 1/t + e^{-\frac{2}{\tau}} \sum_{t=1}^T 1/t + ... \right] \\
&\leq \alpha \left[ \sum_{\theta=0}^T e^{-\frac{\theta}{\tau}} (1 + \ln(T)) \right] \quad \text{since } \sum_{t=1}^T \frac{1}{T} \leq 1 + \ln(T) \\
&\leq \alpha(1 + \ln(T))(1 + \int_0^\infty e^{-\frac{\theta}{\tau}} \, d\theta) \\
&= \alpha(1 + \ln(T))(1 + \tau).
\end{aligned}
$$

$\qquad\square$

We are now ready to prove Theorem 1.

*Proof of Theorem 1.* Combining Eq (3), Lemma 2, Lemma 4 and Lemma 9, we have

$$\sup_{\pi \in \Pi} R(T, \pi)$$

$$\leq (2\tau+2)+\left[\max_{\pi \in \Delta_M} \sum_{t=1}^{T}\langle \mu^{\pi}, r_t\rangle - \sum_{t=1}^{T}\langle \mu^{\pi_t}, r_t\rangle\right] + \left[\sum_{t=1}^{T} 2e^{-\frac{t-1}{\tau}} + \sum_{t=1}^{T}\sum_{\theta=0}^{t-1} e^{-\frac{\theta}{\tau}}\|\mu^{\pi_{t-\theta}} - \mu^{\pi_{t-(\theta+1)}}\|_1\right]$$

$$\leq 4(\tau+1) + \left[\max_{\pi \in \Delta_M} \sum_{t=1}^{T}\langle \mu^{\pi}, r_t\rangle - \sum_{t=1}^{T}\langle \mu^{\pi_t}, r_t\rangle\right] + 2\eta\left(1+\frac{1}{\eta}G_R\right)(1+\ln(T))(1+\tau). \qquad (10)$$

The second term in Eq (10) is bounded by

$$\max_{\pi \in \Delta_M} \sum_{t=1}^{T}\langle \mu^{\pi}, r_t\rangle - \sum_{t=1}^{T}\langle \mu^{\pi_t}, r_t\rangle$$

$$\leq \max_{\pi \in \Delta_{M,\delta}} \sum_{t=1}^{T}\langle \mu^{\pi}, r_t\rangle - \sum_{t=1}^{T}\langle \mu^{\pi_t}, r_t\rangle + 2\delta T\left(|S||A|-1\right) \quad \text{Lemma 8}$$

$$\leq \sum_{t=1}^{T}\langle \mu^{\pi_{t+1}}, r_t\rangle - \sum_{t=1}^{T}\langle \mu^{\pi_t}, r_t\rangle + \frac{T}{\eta} \max_{\mu_1,\mu_2 \in \Delta_{M,\delta}} [R(\mu_1) - R(\mu_2)] + 2\delta T\left(|S||A|-1\right) \text{ Lemma 5}$$

$$\leq \sum_{t=1}^{T}\langle \mu^{\pi_{t+1}}, r_t\rangle - \sum_{t=1}^{T}\langle \mu^{\pi_t}, r_t\rangle + \frac{T}{\eta}\ln(|S||A|) + 2\delta T\left(|S||A|-1\right) \quad \text{by choice of function } R$$

$$\leq \sum_{t=1}^{T}\|r_t\|_{\infty}\|\mu^{\pi_{t+1}} - \mu^{\pi_t}\|_1 + \frac{T}{\eta}\ln(|S||A|) + 2\delta T\left(|S||A|-1\right) \quad \text{by Cauchy-Schwarz inequality}$$

$$\leq \sum_{t=1}^{T}\frac{2\eta}{t}\left(1+\frac{1}{\eta}G_R\right) + \frac{T}{\eta}\ln(|S||A|) + 2\delta T\left(|S||A|-1\right) \quad \text{by Lemma 6}$$

$$\leq 2\eta\left(1+\frac{1}{\eta}G_R\right)(1+\ln(T)) + \frac{T}{\eta}\ln(|S||A|) + 2\delta T\left(|S||A|-1\right).$$

Plugging this result into Eq (10), we get

$$\sup_{\pi \in \Pi} R(T, \pi) \leq 4(\tau+1) + 2\eta(1+\frac{1}{\eta}G_R)(1+\ln(T)) + \frac{T}{\eta}\ln(|S||A|)$$

$$+ 2\delta T\left(|S||A|-1\right) + 2\eta(1+\frac{1}{\eta}G_R)(1+\ln(T))(1+\tau)$$

$$\leq 4(\tau+1) + 4\eta(1+\frac{1}{\eta}G_R)(1+\ln(T))(1+\tau) + \frac{T}{\eta}\ln(|S||A|) + 2\delta T\left(|S||A|-1\right)$$

$$= O\left(\tau + 4\sqrt{\tau T \ln(|S||A|)}\ln(T) + \sqrt{\tau T \ln(|S||A|)} + e^{-\frac{\sqrt{T}}{\sqrt{\tau}}}T|S||A|\right).$$

The proof is completed by choosing $\eta = \sqrt{\frac{T\ln(|S||A|)}{\tau}}$ and $\delta = e^{-\frac{\sqrt{T}}{\sqrt{\tau}}}$, and using the fact that $G_R \leq \max\{|\ln(\delta)|, 1\}$. $\qquad \square$

## B  Proof of Theorem 2

Using Lemma 2 and Lemma 4 in Appendix A, we can obtain a bound on $\Phi$-MDP-Regret as follows.

$$\max_{\pi \in \Pi^\Phi} R(\pi, T) \leq \mathbb{E}\left[(2\tau+2) + \max_{\pi \in \Pi^\Phi}\left[\sum_{t=1}^{T}\rho_t^\pi - \sum_{t=1}^{T}\rho_t\right] + \left[\sum_{t=1}^{T}\rho_t - \mathbb{E}[\sum_{t=1}^{T}r_t(s_t, a_t)]\right]\right]$$

$$= \mathbb{E}\left[(2\tau+2) + \left[\max_{\mu \in \Delta_{M,\delta}^\Phi}\sum_{t=1}^{T}\langle\mu, r_t\rangle - \sum_{t=1}^{T}\langle\mu^{\Phi\tilde{\theta}_t}, r_t\rangle\right] + \left[\sum_{t=1}^{T}\rho_t - \mathbb{E}[\sum_{t=1}^{T}r_t(s_t, a_t)]\right]\right]$$

$$\leq \mathbb{E}\left[2(2\tau+2) + \left[\max_{\mu \in \Delta_{M,\delta}^\Phi}\sum_{t=1}^{T}\langle\mu, r_t\rangle - \sum_{t=1}^{T}\langle\mu^{\Phi\tilde{\theta}_t}, r_t\rangle\right] + \left[\sum_{t=1}^{T}\sum_{i=0}^{t-i}e^{-\frac{i}{\tau}}\|\mu^{\Phi\tilde{\theta}_{t-i}} - \mu^{\Phi\tilde{\theta}_{t-(i+1)}}\|_1\right]\right].$$

Let $\theta_t^*$ be a solution to the following optimization problem:

$$\max_{\theta \in \Theta} \sum_{i=1}^{t-1}\left[\langle\mu, r_i\rangle + \frac{1}{\eta}R^\delta(\mu)\right]$$

$$\text{s.t} \quad \mu = \Phi\theta$$

$$\sum_{s \in S}\sum_{a \in A}\mu(s, a)P(s'|s, a) = \sum_{a \in A}\mu(s', a) \quad \forall s' \in S$$

$$\sum_{s \in S}\sum_{a \in A}\mu(s, a) = 1$$

$$\mu(s, a) \geq 0 \quad \forall s \in S, \forall a \in A.$$

Since $\{\Phi\theta_t^*\}_{t=1}^T$ represents the iterates of RFTL, we can use the regret guarantee of RFTL to bound $\max_{\mu \in \Delta_{M,\delta}^\Phi}\sum_{t=1}^{T}\langle\mu, r_t\rangle - \sum_{t=1}^{T}\langle\mu^{\Phi\theta_t^*}, r_t\rangle$. Notice also that $\mu^{\Phi\theta_t^*} = \Phi\theta_t^*$ as $\theta_t^*$ satisfies all the constraints that ensure $\Phi\theta_t^*$ is an occupancy measure.

In the remainder of the proof, we want to show that the occupancy measures $\mu^{\Phi\tilde{\theta}_t}$ induced by our algorithm's iterates $\Phi\tilde{\theta}_t$ are close to $\mu^{\Phi\theta_t^*}$. The rest of the analysis is to prove that $\|\mu^{\Phi\theta_t^*} - \mu^{\Phi\tilde{\theta}_t}\|_1$ is small. Notice that using the triangle inequality, we can upper bound this distance by

$$\|\mu^{\Phi\theta_t^*} - \mu^{\Phi\tilde{\theta}_t}\|_1 \leq \|\mu^{\Phi\theta_t^*} - P_{\Delta_{M,\delta}^\Phi}(\Phi\tilde{\theta}_t)\|_1 + \|P_{\Delta_{M,\delta}^\Phi}(\Phi\tilde{\theta}_t) - \Phi\tilde{\theta}_t\|_1 + \|\Phi\tilde{\theta}_t - \mu^{\Phi\tilde{\theta}_t}\|_1$$

$$= \|\Phi\theta_t^* - P_{\Delta_{M,\delta}^\Phi}(\Phi\tilde{\theta}_t)\|_1 + \|P_{\Delta_{M,\delta}^\Phi}(\Phi\tilde{\theta}_t) - \Phi\tilde{\theta}_t\|_1 + \|\Phi\tilde{\theta}_t - \mu^{\Phi\tilde{\theta}_t}\|_1.$$

To bound the last term, the following lemma from [3] will be useful. It relates a vector $\Phi\tilde{\theta}$ which is almost feasible with its occupancy measure.

**Lemma 10.** *[Lemma 2 in [3]] Let $u \in \mathbb{R}^{|S||A|}$ be a vector. Let $\mathcal{N}$ be the set of entries $(s, a)$ where $u(s, a) \leq 0$. Assume*

$$\sum_{(s,a)}u(s, a) = 1, \quad \sum_{(s,a) \in \mathcal{N}}|u(s, a)| \leq \epsilon', \quad \|u^\top(P - B)\|_1 \leq \epsilon''.$$

*Vector $[u]_+/\|[u]_+\|_1$ defines a policy, which in turn defines a stationary distribution $\mu^u$. It holds that*

$$\|\mu^u - u\|_1 \leq \tau\ln(\frac{1}{\epsilon'})(2\epsilon' + \epsilon'') + 3\epsilon'.$$

Suppose we are given a vector $\Phi\tilde{\theta}_t$ such that $\|[\Phi\tilde{\theta}_t]_{(\delta,-)}\|_1 \leq \epsilon'$ and $\|(\Phi\tilde{\theta}_t)^\top(P - B)\|_1 \leq \epsilon''$. In view of Lemma 10 and the fact that $\|[\Phi\tilde{\theta}_t]_-\|_1 \leq \|[\Phi\tilde{\theta}_t]_{(\delta,-)}\|_1 \leq \epsilon'$, we have a bound on $\|\Phi\tilde{\theta}_t - \mu^{\Phi\tilde{\theta}_t}\|_1$. The next lemma shows that we can also obtain a bound on $\|P_{\Delta_{M,\delta}^\Phi}(\Phi\tilde{\theta}_t) - \Phi\tilde{\theta}_t\|_1$.

**Lemma 11.** *Let $\Phi\tilde{\theta}_t$ be a vector such that $\|[\Phi\tilde{\theta}]_{(\delta,-)}\|_1 \leq \epsilon'$ and $\|(\Phi\tilde{\theta})^\top(P - B)\|_1 \leq \epsilon''$ for some $\epsilon', \epsilon'' \geq 0$. It holds that*

$$\|P_{\Delta_{M,\delta}^\Phi}(\Phi\tilde{\theta}_t) - \Phi\tilde{\theta}_t\|_1 \leq c(\epsilon' + \epsilon''),$$

*where $c$ is a bound on the $l_\infty$ norm of the Lagrange multipliers of certain linear programming problem.*

*Proof.* The idea comes from sensitivity analysis in Linear Programming (LP) (see for example [36]). Consider the $l_1$ projection problem of $\Phi\tilde{\theta}_t$ onto the set of occupancy measures parametrized by $\Phi$

$$\min_{\theta}\|\mu - \Phi\tilde{\theta}\|_1$$

$$\text{s.t}\quad \mu = \Phi\theta$$
$$\mu^\top 1 = 1$$
$$\mu \geq \delta$$
$$\mu^\top(P - B) = 0$$
$$\theta \in \Theta.$$

It can be reformulated as the following LP

Primal 1: $$\min_{\theta,u}\sum_{(s,a)}u(s,a)$$

$$\text{s.t}\quad u(s,a) - [\Phi\theta](s,a) \geq -[\Phi\tilde{\theta}](s,a)$$
$$u(s,a) + [\Phi\theta](s,a) \geq [\Phi\tilde{\theta}](s,a)$$
$$\mu = \Phi\theta$$
$$\mu^\top 1 = 1$$
$$\mu \geq \delta$$
$$\mu^\top(P - B) = 0$$
$$-\theta(i) \geq -W \quad \forall i = 1,...,d$$
$$\theta(i) \geq 0 \quad \forall i = 1,...,d$$

Let us now consider the perturbed problem 'Primal 2' which arises by perturbing the right hand side vector of 'Primal 1':

Primal 2: $$\min_{\theta,u}\sum_{(s,a)}u(s,a)$$

$$\text{s.t}\quad u(s,a) - [\Phi\theta](s,a) \geq -[\Phi\tilde{\theta}](s,a)$$
$$u(s,a) + [\Phi\theta](s,a) \geq [\Phi\tilde{\theta}](s,a)$$
$$\mu = \Phi\theta$$
$$\mu^\top 1 = 1$$
$$\mu \geq \delta + \tilde{a}$$
$$\mu^\top(P - B) = \tilde{b}$$
$$-\theta(i) \geq -W \quad \forall i = 1,...,d$$
$$\theta(i) \geq 0 \quad \forall i = 1,...,d$$

We choose perturbation vectors $\tilde{a}, \tilde{b}$ such that the optimal value of ' Primal 2' is zero is 0. Let $b$ be the right hand side vector of 'Primal 1' and $b' \triangleq b - \xi$ be that of 'Primal 2' for some vector $\xi$. Since by assumption we have that $\|[\Phi\tilde{\theta}]_{(\delta,-)}\|_1 \leq \epsilon'$ and $\|(\Phi\tilde{\theta})^\top(P - B)\|_1 \leq \epsilon''$ then it holds that $\|b - b'\|_1 = \|\xi\|_1 \leq \epsilon' + \epsilon''$. Let 'Opt. Primal 1' and 'Opt. Primal 2' be the optimal value of the respective problems ('Opt. Primal 2' = 0 by construction) and let $\lambda^*$ be the vector of optimal dual variables of 'Dual 1', the problem dual to 'Primal 1'. Since by assumption, the feasible set of 'Primal 1' is feasible, then the absolute value of the entries of $\lambda^*$ is bounded by some constant $c$.

Now, since $\lambda^*$ is feasible for 'Dual 2', the following sequence of inequalities hold:

$$\text{'Opt. Primal 2'} \geq (\lambda^*)^\top(b - \xi)$$
$$\Longleftrightarrow \text{'Opt. Primal 2'} \geq \text{'Opt. Primal 1'} - (\lambda^*)^\top\xi.$$

Therefore,

$$\text{'Opt. Primal 1'} \leq \text{'Opt. Primal 2'} + \|\lambda^*\|_\infty \|\xi\|_\infty$$
$$= 0 + \|\lambda^*\|_\infty \|\xi\|_1$$
$$\leq c(\epsilon' + \epsilon''),$$

which yields the result. $\square$

Now, we proceed to bound $\|\Phi\theta_t^* - P_{\Delta_{M,\delta}^\Phi}(\Phi\tilde{\theta}_t)\|_1$. Consider the function

$$F_t(\Phi\theta) \triangleq \sum_{i=1}^t [\langle r_i, \Phi\theta \rangle - \frac{1}{\eta} R^\delta(\Phi\theta)]. \tag{11}$$

Since $R^\delta$ is strongly convex over $\Delta_{M,\delta}^\Phi$ with respect to $\|\cdot\|_1$ (but not everywhere over the reals as the extension uses a linear function), we have that $F_t$ is $\frac{t}{\eta}$-strongly concave with respect to $\|\cdot\|_1$ over $\Delta_{M,\delta}^\Phi$. With this in mind, we can prove the following result.

**Lemma 12.** *Let* $\Phi\tilde{\theta}_{t+1}$ *be a vector such that* $\|[\Phi\tilde{\theta}_{t+1}]_{(\delta,-)}\|_1 \leq \epsilon'$ *and* $\|(\Phi\tilde{\theta}_{t+1})^\top(P-B)\|_1 \leq \epsilon''$ *for some* $\epsilon', \epsilon'' \geq 0$. *Let* $\epsilon'''$ *be such that* $F_t(\Phi\theta_{t+1}^*) - F_t(\Phi\tilde{\theta}_{t+1}) \leq \epsilon'''$. *And let* $G_{F_t}$ *be the Lipschitz constant of* $F_t$ *with respect to norm* $\|\cdot\|_1$ *over the set* $\Delta_{M,\delta}^\Phi$. *It holds that*

$$\|\Phi\theta_{t+1}^* - P_{\Delta_{M,\delta}^\phi}(\Phi\tilde{\theta}_{t+1})\|_1 \leq \sqrt{\frac{2\eta}{t}(\epsilon''' + G_{F_t}c(\epsilon' + \epsilon''))}.$$

*Proof.* Since $F_t$ is $\frac{t}{\eta}$-strongly concave over $\Delta_{M,\delta}^\Phi$ and $\Phi\theta_{t+1}^*$ is the optimizer of $F_t$ over $\Delta_{M,\delta}^\Phi$. It holds that

$$\frac{t}{2\eta}\|\Phi\theta_{t+1}^* - \Phi\tilde{\theta}_{t+1}\|_1^2 \leq F_t(\Phi\theta_{t+1}^*) - F_t(P_{\Delta_{M,\delta}^\Phi}(\Phi\tilde{\theta}_{t+1}))$$
$$\leq F_t(\Phi\theta_{t+1}^*) - F_t(\Phi\tilde{\theta}_{t+1}) + G_{F_t}\|P_{\Delta_{M,\delta}^\Phi}(\Phi\tilde{\theta}_{t+1}) - \Phi\tilde{\theta}_{t+1}\|_1$$
$$\leq \epsilon''' + G_{F_t}\|P_{\Delta_{M,\delta}^\Phi}(\Phi\tilde{\theta}_{t+1}) - \Phi\tilde{\theta}_{t+1}\|_1 \quad \text{by assumption}$$
$$\leq \epsilon''' + G_{F_t}c(\epsilon' + \epsilon'') \quad \text{by Lemma 11}$$

which yields the result. $\square$

The next lemma bounds the Lipschitz constant $G_{F_t}$.

**Lemma 13.** *Let* $\eta = \sqrt{\frac{T}{\tau}}$, $\delta = e^{-\sqrt{T}}$. *The function* $F_t(\mu) : \mathbb{R}^{|S||A|} \to \mathbb{R}$ *is* $G_{F_t}$*-Lipschitz continuous on variables* $\mu$ *with respect to norm* $\|\cdot\|_1$ *over* $\Delta_{M,\delta}^\Phi$ *with* $G_{F_t} \leq t(1 + 2\sqrt{\tau}\ln(dW))$.

*Proof.* It suffices to find a an upper bound for $\|\nabla_\mu F_t(\mu)\|_\infty$. Since $\nabla_\mu F_t(\mu) = \sum_{i=1}^t r_i - \frac{t}{\eta}\nabla_\mu R^\delta(\mu)$, we have that

$$\|\nabla_\mu F_t(\mu)\|_\infty \leq \|\sum_{i=1}^t r_i\|_\infty + \frac{t}{\eta}\|\nabla_\mu R^\delta(\mu)\|_\infty \quad \text{by triangle inequality}$$
$$\leq t + \frac{t}{\eta}\|\nabla_\mu R^\delta(\mu)\|_\infty \quad \text{since } |r_i(s,a)| \leq 1$$
$$\leq t + \frac{t}{\eta}\max\{|1 + \ln(\delta)|, |1 + \ln(dW)|\} \quad \text{as in the proof of Lemma 7 .}$$

The second to last inequality holds since $|\frac{d}{dx}x\ln(x)| = |1 + \ln(x)|$ and the maximum will occur at $x = \delta$ or $x = [\Phi\theta](s,a)$, $[\Phi\theta](s,a)$ can be bounded by $Wd$. Plugging in the values for $\eta$ and $\delta$ we

get

$$\|\nabla_\mu F_t(\mu)\|_\infty \le t + \frac{t\tau}{\sqrt{T}}(1 + \max\{\sqrt{T}, \ln(dW)\})$$

$$\le t + \frac{t\tau}{\sqrt{T}}(2\sqrt{T}\ln(dW))$$

$$= t(1 + 2\sqrt{\tau}\ln(dW)).$$

$\square$

Combining the previous three lemmas, we obtain the following result.

**Lemma 14.** *Let $\Phi\tilde{\theta}_{t+1}$ be a vector such that $\|[\Phi\tilde{\theta}_{t+1}]_{(\delta,-)}\|_1 \le \epsilon'$ and $\|(\Phi\tilde{\theta}_{t+1})^\top(P-B)\|_1 \le \epsilon''$ for some $\epsilon', \epsilon'' \ge 0$. Let $\epsilon'''$ be such that $F_t(\Phi\theta^*_{t+1}) - F_t(\Phi\tilde{\theta}_{t+1}) \le \epsilon'''$. And let $G_{F_t}$ be the Lipschitz constant of $F_t$ with respect to norm $\|\cdot\|_1$ over the set $\Delta^\Phi_{M,\delta}$. It holds that*

$$\|\mu^{\Phi\theta^*_t} - \mu^{\Phi\tilde{\theta}_t}\|_1 \le \tau \ln(\frac{1}{\epsilon'})(2\epsilon' + \epsilon'') + 3\epsilon' + c(\epsilon' + \epsilon'') + \sqrt{\frac{2\eta}{t}(\epsilon''' + G_{F_t}c(\epsilon' + \epsilon''))}.$$

*Proof.* By triangle inequality, we have

$$\|\mu^{\Phi\theta^*_t} - \mu^{\Phi\tilde{\theta}_t}\|_1 \le \|\Phi\theta^*_t - P_{\Delta^\Phi_{M,\delta}}(\Phi\tilde{\theta}_t)\|_1 + \|P_{\Delta^\Phi_{M,\delta}}(\Phi\tilde{\theta}_t) - \Phi\tilde{\theta}_t\|_1 + \|\Phi\tilde{\theta}_t - \mu^{\Phi\tilde{\theta}_t}\|_1.$$

Using Lemmas 10, 11, and 12 to bound the first, second, and third terms respectively yields the result.

$\square$

Now we can upper bound the bound on the $\Phi$-MDP-Regret, Eq (11), using triangle inequality and Lemma 14. For the bound to be useful we want to be able to produce vectors $\{\Phi\tilde{\theta}_t\}_{t=1}^T$ that satisfy the conditions of Lemma 14 with $\epsilon', \epsilon'', \epsilon'''$ that are small enough. It is also important that we produce $\{\Phi\tilde{\theta}_t\}_{t=1}^T$ in a computationally efficient manner. At time $t$, our approach to generate $\Phi\tilde{\theta}_t$, will be to run Projected Stochastic Gradient Descent on function 7. The following theorem from [3] will be useful.

**Theorem 3** (Theorem 3 in [3]). *Let $\mathcal{Z} \subset \mathbb{R}$ be a convex set such that $\|z\|_2 \le Z$ for all $z \in \mathcal{Z}$ for some $Z > 0$. Let $f$ be a concave function defined over $\mathcal{Z}$. Let $\{z_k\}_{k=1}^K \in \mathcal{Z}^T$ be the iterates of Projected Stochastic Gradient Ascent, i.e. $z_{k+1} \leftarrow P_\mathcal{Z}(x_k + \eta f'_t)$ where $P_\mathcal{Z}$ is the euclidean projection onto $\mathcal{Z}$, $\eta$ is the step-size and $\{f'_k\}_{k=1}^K$ are such that $\mathbb{E}[f'_k|z_k] = \nabla f(z_k)$ with $\|f'_k\|_2 \le F$ for some $F > 0$. Then, for $\eta = \frac{Z}{(F\sqrt{K})}$ for all $\kappa \in (0,1)$, with probability at least $1 - \kappa$ it holds that*

$$\max_{z\in\mathcal{Z}} f(z) - f(\frac{1}{K}\sum_{k=1}^K z_k) \le \frac{ZF}{\sqrt{K}} + \sqrt{\frac{(1 + 4Z^2 K)\left(2\ln(\frac{1}{\kappa}) + d\ln(1 + \frac{Z^2 K}{d})\right)}{K^2}}.$$

In view of Theorem 3 we need to design a stochastic subgradient for $c^{t,\eta}$ and a bound for its $l$-2 norm. We follow the approach in [3], we notice however that the objective function considered in [3] does not contain the regularizer $R^\delta$ so must take care of that in our analysis.

Lemma 1 creates a stochastic subgradient for $c^{t,\eta}$ and provides an upper bound for its $l$-2 norm. We now present its proof.

*Proof of Lemma 1.* Let us first compute $\nabla_\theta c^{\eta,t}(\theta)$. Define $r_{:t} \triangleq \sum_{i=1}^t r_i$ By definition we have

$$c^{\eta,t}(\theta) = (\Phi\theta)^\top r_{:t} - \frac{t}{\eta}\sum_{(s,a)} R^\delta_{(s,a)}(\Phi\theta) - H_t\|[\Phi\theta]_{(\delta,-)}\|_1 - H_t\|(P-B)^\top\Phi\theta\|_1$$

$$= \theta^\top(\Phi^\top r_{:t}) - \frac{t}{\eta}\sum_{(s,a)} R^\delta_{(s,a)}(\Phi\theta) - H_t\sum_{(s,a)}[\Phi_{(s,a),:}\theta]_{(\delta,-)} - H_t\sum_s |[(P-B)^\top\Phi]_{s,:}\theta|.$$

So, we get

$$\nabla_\theta c^{t,\eta}(\theta) = \Phi^\top r_{:t} - \frac{t}{\eta} \sum_{(s,a)} \nabla_\theta R^\delta_{(s,a)}(\Phi\theta)$$

$$- H_t \sum_{(s,a)} -\Phi_{(s,a),:} \mathbb{I}\{\Phi_{(s,a),:}\theta \leq \delta\} - H_t \sum_s [(P-B)^\top \Phi]_{s,:} sign([(P-B)^\top \Phi]_{s,:}\theta).$$

We design a stochastic subgradient $g$ of $\nabla_\theta c^{\eta,t}(\theta)$ by sampling a state-action pair $(s',a')$ from the given distribution $q_1$ and a state $s''$ from distribution $q_2$. Then, we have

$$g_{s',a',s''}(\theta) = \Phi^\top r_{:t} + \frac{H_t}{q_1(s',a')}\Phi_{(s',a'),:}\mathbb{I}\{\Phi_{(s',a'),:} \leq \delta\}$$

$$- \frac{H_t}{q_2(s'')}[(P-B)^\top \Phi]_{s'',:} sign([(P-B)^\top \Phi]_{s'',:}\theta) - \frac{t}{\eta q_1(s',a')}\nabla_\theta R^\delta_{(s',a')}(\Phi\theta).$$

We will also give a closed form expression of $\nabla_\theta R^\delta_{(s',a')}(\Phi\theta)$ in the proof below. By construction, it holds that $\mathbb{E}_{(s',a')\sim q_1, s''\sim q_2}[g_{s',a',s''}(\theta)|\theta] = \nabla_\theta c^{t,\eta}(\theta)$. To simplify notation, let $g(\theta) = g_{s',a',s''}(\theta)$.

We now bound $\|g(\theta)\|_2$ with probability 1. First, we have

$$\|\Phi^\top r_{:t}\|_2 = \sqrt{\sum_{i=1}^d (r_{:t}^\top \Phi_{:,i})^2}$$

$$\leq \sqrt{\sum_{i=1}^d (\|r_{:t}\|_\infty \|\Phi_{:,i}\|_1)^2} \quad \text{by Cauchy-Schwarz}$$

$$\leq \sqrt{dt^2 1} = t\sqrt{d},$$

where the last inequality holds since $\|r_i\|_\infty \leq 1$ for $t = 1, ..., T$ and each column of $\Phi$ is a probability distribution. Next, we have

$$\left\| \frac{H_t}{q_1(s',a')}\Phi_{(s',a'),:}\mathbb{I}\{\Phi_{(s',a'),:}\theta \leq \delta\} \right\|_2 \leq H_t C_1, \quad \text{and}$$

$$\left\| -\frac{H_t}{q_2(s'')}[(P-B)^\top \Phi]_{s'',:} sign([(P-B)^\top \Phi]_{s'',:}\theta) \right\|_2 \leq H_t C_2,$$

where $C_1$ and $C_2$ are defined in (9). Finally, we bound $\|\nabla_\theta R^\delta_{(s,a)}(\Phi\theta)\|_2$. By definition of $R^\delta_{(s,a)}$ in Eq 8, we need to compute the gradients of the negative entropy function $\nabla_\theta R(\Phi\theta)$. Let us compute $\frac{d}{d\theta_i}R(\Phi\theta)$ as follows.

$$\frac{d}{d\theta_i}R(\Phi\theta) = \sum_{(s,a)} \frac{d}{d\theta_i}R_{(s,a)}(\Phi\theta)$$

$$= \sum_{(s,a)} \frac{d}{d\theta_i}\left[ (\sum_{k=1}^d \Phi_{(s,a),k}\theta_k)\ln(\sum_{k=1}^d \Phi_{(s,a),k}\theta_k) \right]$$

$$= \sum_{(s,a)}(\sum_{k=1}^d \Phi_{(s,a),k}\theta_k)(\frac{d}{d\theta_i}\ln(\sum_{k=1}^d \Phi_{(s,a),k}\theta_k)) + \ln(\sum_{k=1}^d \Phi_{(s,a),k}\theta_k)\Phi_{(s,a),i}$$

$$= \sum_{(s,a)}(\sum_{k=1}^d \Phi_{(s,a)}\theta_k)\frac{1}{\sum_{k=1}^d \Phi_{(s,a)}\theta_k}\frac{d}{d\theta_i}(\sum_{k=1}^d \Phi_{(s,a),k}\theta_k) + \ln(\sum_{k=1}^d \Phi_{(s,a),k}\theta_k)\Phi_{(s,a),i}$$

$$= \sum_{(s,a)}\Phi_{(s,a),i} + \ln(\sum_{k=1}^d \Phi_{(s,a),k}\theta_k)\Phi_{(s,a),i}.$$

We are also interested in the gradient of the linear extension of $R_{(s,a)}(x)$: $R_{(s,a)}(\delta) + \frac{d}{dx}R_{(s,a)}(\delta)(x - \delta)$ which is equal to $\delta \ln(\delta) + (1 + \ln(\delta))(x - \delta)$. So we upper bound $|\frac{d}{d\theta_i}\delta \ln(\delta) + (1 + \ln(\delta))(\Phi_{(s,a),:}\theta - \delta)|$ by

$$|\frac{d}{d\theta_i}\delta \ln(\delta) + (1 + \ln(\delta))(\Phi_{(s,a),:}\theta - \delta)|$$
$$=|\frac{d}{d\theta_i}(1 + \ln(\delta))(\Phi_{(s,a),:}\theta - \delta)|$$
$$=|(1 + \ln(\delta))\Phi_{(s,a),i}|.$$

It follows that

$$\|\nabla_\theta R^\delta_{(s,a)(\Phi\theta)}\|_2$$
$$\leq \left(\sum_{i=1}^d \left[\max\{\Phi_{(s,a),i} + \ln(W\sum_{k=1}^d \Phi_{(s,a),k})\Phi_{(s,a),i}, |(1 + \ln(\delta))\Phi_{(s,a),i}|\}\right]^2\right)^{1/2}$$
$$\leq \left(\sum_{i=1}^d \left[(1 + \max\{\ln(W\sum_{k=1}^d \Phi_{(s,a),k}), |\ln(\delta)|\}\Phi_{(s,a),i}\right]^2\right)^{1/2}$$
$$\leq \left(\sum_{i=1}^d \left[(1 + \max\{\ln(Wd), |\ln(\delta)|\}\Phi_{(s,a),i}\right]^2\right)^{1/2}$$
$$\leq (1 + \ln(Wd) + |\ln(\delta)|)\|\Phi_{(s,a),:}\|_2.$$

Thus, we have $\|\frac{t}{\eta q_1(s',a')}\nabla_\theta R^\delta_{(s',a')}(\Phi\theta)\|_2 \leq \frac{t}{\eta}(1 + \ln(Wd) + |\ln(\delta)|)C_1$. Using triangle inequality, we have that with probability 1

$$\|g(\theta)\|_2 \leq t\sqrt{d} + H(C_1 + C_2) + \frac{t}{\eta}(1 + \ln(Wd) + |\ln(\delta)|)C_1.$$

$\square$

By using Lemma 1, as well as the fact that $\theta \in \Theta$ and $\|\theta\|_2 \leq d\|\theta\|_\infty$ implies $\|\theta\|_2 \leq W$, we can prove the following.

**Lemma 15.** *For all $t = 1, ..., T$, $\eta > 0$, $\kappa \in (0, 1)$, after running $K(t)$ iterations of Projected Stochastic Gradient Ascent on function $c^{\eta,t}(\theta)$ over the set $\Theta^\Phi$ and using step-size $\frac{\sqrt{d}W}{\sqrt{K(t)}G'}$ with $G' = t\sqrt{d} + H_t(C_1 + C_2) + \frac{t}{\eta}(1 + \ln(Wd) + |\ln(\delta)|)C_1$ with probability at least $1 - \kappa$ it holds that*

$$\sum_{i=1}^t \left[\langle r_i, \Phi\theta^*_{t+1}\rangle - \frac{1}{\eta}R^\delta(\Phi\theta^*_{t+1})\right]$$
$$- \left[\sum_{i=1}^t \left[\langle r_i, \Phi\tilde{\theta}_{t+1}\rangle - \frac{1}{\eta}R^\delta(\Phi\tilde{\theta}_{t+1})\right] - H_t\|(\Phi\tilde{\theta}_{t+1})^\top(P - B)\|_1 - H_t\|[\Phi\tilde{\theta}_{t+1}]_{(\delta,-)}\|_1\right]$$
$$\leq \frac{\sqrt{d}WG'}{\sqrt{K(t)}} + \sqrt{\frac{(1 + 4dW^2K(t))(2\ln(\frac{1}{\kappa}) + d\ln(1 + \frac{dW^2K(t)}{d}))}{K(t)^2}}.$$

*Proof.* The proof follows from applying Theorem 3 on function $c^{\eta,t}(\theta)$. Using the bound of the stochastic gradients from Lemma 1, as well as the fact that $\max_{\theta \in \Theta^\Phi} c^{\eta,t}(\theta) \geq c^{\eta,t}(\theta^*_{t+1})$ and since $\Phi\theta^*_{t+1}$ is feasible, we have $\|(\Phi\theta^*_{t+1})^\top(P - B)\|_1 = 0$ and $\|[\Phi\theta^*_{t+1}]_{(\delta,-)}\|_1 = 0$. $\square$

We remark that we did not relax the constraint $(\Phi\theta)^\top 1 = 1$ and in fact when we use Projected Gradient Ascent we are projecting onto a subset of that hyperplane, although $\Phi$ has $|S||A|$ rows we

can precompute the vector $\Phi^\top 1 \in \mathbb{R}^d$ so that all projections to the subset of the hyper plane given by $(\Phi\theta)^\top 1 = 1$ can be done in $O(poly(d))$ time.

The next lemma bounds the largest difference the function $F_t(\Phi\theta)$ can take over $\theta \in \Theta^\Phi$. It will be clear later why this bound is needed.

**Lemma 16.** *For all $t = 1, ..., T$. It holds that*

$$\max_{\theta_1, \theta_2 \in \Theta^\Phi} F_t(\Phi\theta_1) - F_t(\Phi\theta_2) \leq t \left[2 + \frac{1}{\eta} \ln(|S||A|)\right].$$

*Proof.* By definition of $F_t$ it suffices to bound

$$\sum_{i=1}^t \langle r_i, \Phi\theta_1 - \Phi\theta_2 \rangle + \frac{t}{\eta} \left[R^\delta(\Phi\theta_2) - R^\delta(\Phi\theta_1)\right].$$

Now, we have

$$\sum_{i=1}^t \langle r_i, \Phi\theta_1 - \Phi\theta_2 \rangle \leq \sum_{i=1}^t \|r_i\|_\infty \|\Phi\theta_1 - \Phi\theta_2\|_1 \quad \text{By Cauchy-Schwarz}$$

$$\leq \sum_{i=1}^t 1 \|\Phi\theta_1 - \Phi\theta_2\|_1$$

$$\leq \sum_{i=1}^t \|\Phi\theta_1\|_1 + \|\Phi\theta_2\|_1 \quad \text{by triangle inequality}$$

$$\leq 2t,$$

where the last inequality holds since all entries of $\Phi$ and $\theta$ are nonnegative, and $(\Phi\theta)^\top 1 = 1$ for all $\theta \in \Theta^\Phi$.

It is well known that the minimizer of $R(\mu)$ for $\mu \in \Delta^{|S||A|}$ is $-\ln(|S||A|)$. Moreover, its optimal solution $\mu^*$ is equal to the vector with value $1/(|S||A|)$ on each of its entries, which is of course in the interior of the simplex. Notice that since $R^\delta$ is an extension of $R$, if $\delta$ is sufficiently small (which we ensure by the choice of $\delta$ later in the analysis), the minimizer of $R^\delta(\Phi\theta)$ for $\theta \in \Theta^\Phi$ will be bounded below by $-\ln(|S||A|)$. That is

$$-\ln(|S||A|) \leq \min_{\theta \in \Theta^\Phi} R^\delta(\Phi\theta).$$

We upper bound $\max_{\theta \in \Theta^\Phi} R^\delta(\Phi\theta)$. By construction $R^\delta(\Phi\theta) \leq R(\Phi\theta)$ for all $0 \leq \theta \in \Theta^\Phi$. Since $\theta \geq 0$, $1^\top \Phi\theta = 1$ defines the set $\Phi^\Theta$ and $\Phi$ has probability distributions as its columns it holds that $R(\Phi\theta) \leq 0$, thus $R^\delta(\Phi\theta) \leq 0$. We have shown that

$$\sum_{i=1}^t \langle r_i, \Phi\theta_1 - \Phi\theta_2 \rangle + \frac{t}{\eta} \left[R^\delta(\Phi\theta_2) - R^\delta(\Phi\theta_1)\right] \leq 2t + \frac{t}{\eta} \left[\ln(|S||A|)\right]$$

which finishes the proof. $\qquad\square$

Lemma 14 assumes we have at our disposal a vector $\Phi\tilde{\theta}_{t+1}$ such that $\|[\Phi\tilde{\theta}_{t+1}]_{(\delta,-)}\|_1 \leq \epsilon'$ and $\|(\Phi\tilde{\theta}_{t+1})^\top(P - B)\|_1 \leq \epsilon''$, and $F_t(\Phi\theta_{t+1}^*) - F_t(\Phi\tilde{\theta}_{t+1}) \leq \epsilon'''$ for some $\epsilon', \epsilon'', \epsilon''' \geq 0$. We now show how to obtain such error bounds by running at each time step $t$, $K(t)$ iterations of PSGA and using Lemma 15.

**Lemma 17.** *For $t = 1, ..., T$, let $b_{K(t)}$ the right hand side of the equation in the bound of Lemma 15 and assume the same conditions hold. After $K(t)$ iterations of PSGA, with probability at least $1 - \kappa$, it holds that*

$$\|[\Phi\tilde{\theta}_{t+1}]_{(\delta,-)}\|_1 \leq \frac{1}{H_t} \left[b_{K(t)} + t[2 + \frac{1}{\eta}\ln(|S||A|)]\right],$$

$$\|(\Phi\tilde{\theta}_{t+1})^\top(P - B)\|_1 \leq \frac{1}{H_t} \left[b_{K(t)} + t[2 + \frac{1}{\eta}\ln(|S||A|)]\right],$$

$$F_t(\Phi\theta_{t+1}^*) - F_t(\Phi\tilde{\theta}_{t+1}) \leq b_{K(t)}.$$

*Proof.* To show the first two inequalities, notice that Lemma 15 implies

$$H_t\|[\Phi\tilde{\theta}_{t+1}]_{(\delta,-)} + H_t\|(\Phi\tilde{\theta}_{t+1})^\top(P-B)\|_1 \le b_{K(t)} + F_t(\Phi\tilde{\theta}_{t+1}) - F_t(\Phi\theta^*_{t+1})$$

$$\le b_{K(t)} + t\left[2 + \frac{1}{\eta}\ln(|S||A|)\right],$$

where the last inequality holds by Lemma 16. Since $\|\cdot\|_1 \ge 0$ we get the desired results. To show that $F_t(\Phi\theta^*_{t+1}) - F_t(\Phi\tilde{\theta}_{t+1}) \le b_{K(t)}$ again use Lemma 15 and the fact that $\|\cdot\|_1 \ge 0$. $\qquad\square$

We are ready to prove the main theorem from this section.

*Proof of Theorem 2.* Recall the $\Phi$-MDP-Regret regret bound from Equation 11.

$$\max_{\pi\in\Pi^\Phi} R(\pi,T)$$

$$\le \mathbb{E}_{PSGA}[(4\tau+4)$$

$$+ [\max_{\mu\in\Delta^\Phi_{M,\delta}}\sum_{t=1}^T\langle\mu,r_t\rangle - \sum_{t=1}^T\langle\mu^{\Phi\tilde{\theta}_t},r_t\rangle] + [\sum_{t=1}^T\sum_{i=0}^{t-i}e^{-\frac{i}{\tau}}\|\mu^{\Phi\tilde{\theta}_{t-i}} - \mu^{\Phi\tilde{\theta}_{t-(i+1)}}\|_1]].$$

Since it is cumbersome to work with the $\mathbb{E}_{PSGA}[\cdot]$ in our bounds let us make the following argument. For $t=1,...,T$, define $\mathcal{E}_t$ be the event that the inequality in Lemma 15 holds, let $\mathcal{E} \triangleq \cap_{t=1}^T\mathcal{E}_t$. For any random variable $X$ we know that $\mathbb{E}_{PSGA}[X] = \mathbb{E}_{PSGA}[X|\mathcal{E}]P(\mathcal{E}) + \mathbb{E}_{PSGA}[X|\mathcal{E}^c]P(\mathcal{E}^c)$. Let us work conditioned on the event $\mathcal{E}$, we will later bound $\mathbb{E}_{PSGA}[X|\mathcal{E}^c]P(\mathcal{E}^c)$.

By triangle inequality, Cauchy-Schwarz inequality, and the fact $\|r_t\|_\infty \le 1$ for $t=1,...,T$, it holds that

$$\max_{\mu\in\Delta^\Phi_{M,\delta}}\sum_{t=1}^T\langle\mu,r_t\rangle - \sum_{t=1}^T\langle\mu^{\Phi\tilde{\theta}_t},r_t\rangle \le \max_{\mu\in\Delta^\Phi_{M,\delta}}\sum_{t=1}^T\langle\mu,r_t\rangle - \sum_{t=1}^T\langle\mu^{\Phi\theta^*_t},r_t\rangle + \sum_{t=1}^T\|\mu^{\Phi\theta^*_t} - \mu^{\Phi\tilde{\theta}_t}\|_1.$$

Notice that

$$\|\mu^{\Phi\tilde{\theta}_{t-i}} - \mu^{\Phi\tilde{\theta}_{t-(i+1)}}\|_1$$

$$\le\|\mu^{\Phi\theta^*_{t-i}} - \mu^{\Phi\theta^*_{t-(i+1)}}\|_1 + \|\mu^{\Phi\tilde{\theta}_{t-i}} - \mu^{\Phi\theta^*_{t-i}}\|_1 + \|\mu^{\Phi\tilde{\theta}_{t-(i+1)}} - \mu^{\Phi\theta^*_{t-(i+1)}}\|_1.$$

Therefore, we have

$$\max_{\pi\in\Pi^\Phi} R(\pi,T)$$

$$\le 2(2\tau+2) + \left[\max_{\mu\in\Delta^\Phi_{M,\delta}}\sum_{t=1}^T\langle\mu,r_t\rangle - \sum_{t=1}^T\langle\mu^{\Phi\theta^*_t},r_t\rangle\right] + \left[\sum_{t=1}^T\sum_{i=0}^{t-i}e^{-\frac{i}{\tau}}\|\mu^{\Phi\theta^*_{t-i}} - \mu^{\Phi\theta^*_{t-(i+1)}}\|_1\right]$$

$$+ \sum_{t=1}^T\|\mu^{\Phi\theta^*_t} - \mu^{\Phi\tilde{\theta}_t}\|_1 + \sum_{t=1}^T\sum_{i=0}^{t-i}e^{-\frac{i}{\tau}}\left(\|\mu^{\Phi\tilde{\theta}_{t-i}} - \mu^{\Phi\theta^*_{t-i}}\|_1 + \|\mu^{\Phi\tilde{\theta}_{t-(i+1)}} - \mu^{\Phi\theta^*_{t-(i+1)}}\|_1\right)$$

$$\le O\left(\tau + 4\sqrt{\tau T}\ln(T) + \sqrt{\tau T}\ln(|S||A|) + e^{-\sqrt{T}}T|S||A|\right)$$

$$+ \sum_{t=1}^T\|\mu^{\Phi\theta^*_t} - \mu^{\Phi\tilde{\theta}_t}\|_1 + \sum_{t=1}^T\sum_{i=0}^{t-i}e^{-\frac{i}{\tau}}\left(\|\mu^{\Phi\tilde{\theta}_{t-i}} - \mu^{\Phi\theta^*_{t-i}}\|_1 + \|\mu^{\Phi\tilde{\theta}_{t-(i+1)}} - \mu^{\Phi\theta^*_{t-(i+1)}}\|_1\right),$$

where the second inequality follows from the proof of Theorem 1 since we chose the same parameters $\eta = \sqrt{\frac{T}{\tau}}, \delta = e^{-\sqrt{T}}$.

If we choose $K(t)$ such that $\|\mu^{\Phi\theta_t^*} - \mu^{\Phi\tilde{\theta}_t}\|_1$ are less than or equal to a constant $\epsilon(\epsilon_t', \epsilon_t'', \epsilon_t''', K(t))$ for all $t = 1, ..., T$ we have

$$\sum_{t=1}^{T} \|\mu^{\Phi\theta_t^*} - \mu^{\Phi\tilde{\theta}_t}\|_1 + \sum_{t=1}^{T}\sum_{i=0}^{t-i} e^{-\frac{i}{\tau}}\left( \|\mu^{\Phi\tilde{\theta}_{t-i}} - \mu^{\Phi\theta_{t-i}^*}\|_1 + \|\mu^{\Phi\tilde{\theta}_{t-(i+1)}} - \mu^{\Phi\theta_{t-(i+1)}^*}\|_1 \right)$$

$$\leq T\epsilon + 2T\epsilon(1 + \int_0^\infty e^{-\frac{x}{\tau}}dx)$$

$$\leq T\epsilon + 2T\epsilon(1 + \tau)$$

$$= T(1 + 2(1 + \tau))\epsilon.$$

We have that

$$\max_{\pi \in \Pi^\Phi} R(\pi, T) \leq O\left( \tau + 4\sqrt{\tau T}\ln(T) + \sqrt{\tau T}\ln(|S||A|) + e^{-\sqrt{T}}T|S||A| + T\tau\epsilon \right).$$

Let $\epsilon_t' = \epsilon_t'' = \frac{1}{H_t}\left[ b_{K(t)} + t[2 + \frac{1}{\eta}\ln(|S||A|)] \right]$, $\epsilon_t''' = b_{K(t)}$. By Lemma 14, we have that

$$\epsilon \leq \tau\ln(\frac{1}{\epsilon'})(2\epsilon' + \epsilon'') + 3\epsilon' + c(\epsilon' + \epsilon'') + \sqrt{\frac{2\eta}{t}(\epsilon''' + G_{F_t}c(\epsilon' + \epsilon''))}.$$

By Lemma 13 we know that $G_{F_t} \leq t(1 + 2\sqrt{\tau}\ln(dW))$ so that

$$\epsilon \leq \tau\ln(\frac{1}{\epsilon'})(2\epsilon' + \epsilon'') + 3\epsilon' + c(\epsilon' + \epsilon'') + \sqrt{\frac{2\sqrt{T}}{\sqrt{\tau}}(\epsilon''' + c[1 + 2\sqrt{\tau}\ln(dW)](\epsilon' + \epsilon''))},$$

where we plugged in the value for $\eta$. It is easy to see that the right hand side of the last inequality bounded above by $O(\tau\ln(\frac{1}{\epsilon'})T^{1/4}c\sqrt{dW}(\epsilon' + \epsilon'' + \epsilon'''))$. So that forcing all $\epsilon', \epsilon'', \epsilon'''$ to be $O(\frac{1}{\sqrt{dW}\tau^{3/2}T^{3/4}})$ will ensure $T\tau\epsilon$ to be $O(c\sqrt{\tau T})$ ensuring that $\max_{\pi \in \Pi^\Phi} R(\pi, T) \leq O(c\sqrt{\tau T}\ln(T)\ln(|S||A|))$.

Since $\epsilon' = \epsilon'' = \frac{1}{H_t}b_{K(t)} + \frac{1}{H_t}t2 + \frac{1}{H_t}t\frac{\sqrt{\tau}}{\sqrt{T}}\ln(|S||A|)$ we choose $H_t = \sqrt{dW}t\tau^2 T^{3/4}$, this ensures that $\frac{1}{H_t}t2 + \frac{1}{H_t}t\frac{\sqrt{\tau}}{\sqrt{T}}\ln(|S||A|)$ are bounded above by $O(\frac{1}{\sqrt{dW}\tau^{3/2}T^{3/4}})$. We now must choose $K(t)$ so that $\frac{1}{H_t}b_{K(t)}$ and $\epsilon_t'''$ are both $O(\frac{1}{\sqrt{dW}\tau^{3/2}T^{3/4}})$. Since by the choice of $H_t$ we have $\frac{1}{H_t}b_{K(t)} \leq b_{K(t)}$ it suffices to bound $b_{K(t)}$.

Set $\kappa = \frac{1}{T^2}$ in Lemma 15 and recall we are working conditioned on $\mathcal{E}$, we have that for all $t = 1, ..., T$

$$b_{K(t)} = \frac{\sqrt{dW}t\sqrt{d} + H_t(C_1 + C_2) + \frac{t}{\eta}(1 + \ln(Wd) + |\ln(\delta)|)C_1}{\sqrt{K(t)}}$$

$$+ \sqrt{\frac{(1 + 4dW^2 K(t))(2\ln(\frac{1}{\kappa}) + d\ln(1 + \frac{dW^2 K(t)}{d}))}{K(t)^2}}$$

$$\leq O\left( \frac{WtdH_t(C_1 + C_2)\sqrt{T}\sqrt{\tau}\ln(WTd)}{\sqrt{T}\sqrt{K(t)}} \right)$$

$$= O\left( \frac{Wt^2 d(C_1 + C_2)\tau^{5/2}T^{3/4}\ln(WTd)}{\sqrt{K(t)}} \right).$$

Setting

$$\frac{Wt^2 d(C_1 + C_2)\tau^{5/2}T^{3/4}\ln(WTd)}{\sqrt{K(t)}} = \frac{1}{\sqrt{dW}\tau^{3/2}T^{3/4}}$$

and solving for $K(t)$, we get that $K(t) = \left[ W^{3/2}t^2 d^{3/2}\tau^4(C_1 + C_2)T^{3/2}\ln(WTd) \right]^2$, which ensures $b_{K(t)} = O(\frac{1}{\sqrt{dW}\tau^{3/2}T^{3/4}})$.

By the choice of $\kappa$ in Lemma 15, we have that for each $t = 1, ..., T$ with probability at least $1 - \frac{1}{T^2}$, $\|\mu^{\Phi\theta_t^*} - \mu^{\Phi\tilde{\theta}_t}\|_1 \leq O(\sqrt{dW}\frac{1}{\tau^{3/2}T^{3/4}})$. This implies that

$\Phi$-MDP-Regret

$$\leq O(c\sqrt{\tau T}\ln(T)\ln(|S||A|))P(\mathcal{E})$$

$$+ \left[O\left(\tau + 4\sqrt{\tau T}\ln(T) + \sqrt{\tau T}\ln(|S||A|) + e^{-\sqrt{T}}T|S||A| + \sum_{t=1}^{T}\|\mu^{\Phi\theta_t^*} - \mu^{\Phi\tilde{\theta}_t}\|_1\right)\right]P(\mathcal{E}^c)$$

Notice that since $\mu^{\Phi\theta_t^*}$, and $\mu^{\Phi\tilde{\theta}_t}$ are probability distributions then $\|\mu^{\Phi\theta_t^*} - \mu^{\Phi\tilde{\theta}_t}\|_1 \leq 2$. So that

$$\Phi\text{-MDP-Regret} \leq O(c\sqrt{\tau T}\ln(T)\ln(|S||A|)) + O(T)P(\mathcal{E}^c)$$

where we upper bounded $P(\mathcal{E})$ with 1. Notice that by the choice of $\kappa$, $P(\mathcal{E}^c) = P(\cup_{t=1}^{T}\mathcal{E}_i^c) \leq \sum_{t=1}^{T}P(\mathcal{E}_t^c) \leq \frac{1}{T}$ so that $O(T)P(\mathcal{E}^c) = O(1)$. This completes the proof.

$\square$

## C  Bounding the problem dependent constant in Theorem 2

Consider the LP formulation of the $l_1$ projection problem of $\Phi\tilde{\theta}$ onto $\Delta_{M,\delta}^{\Phi}$.

$$\min_{\theta,u} \sum_{(s,a)} u(s,a)$$

$$\text{s.t}\quad u(s,a) - [\Phi\theta](s,a) \geq -[\Phi\tilde{\theta}](s,a), \quad u(s,a) + [\Phi\theta](s,a) \geq [\Phi\tilde{\theta}](s,a),$$

$$\mu = \Phi\theta, \quad \mu^\top 1 = 1, \quad \mu \geq \delta, \quad \mu^\top(P-B) \geq 0, \quad -\mu^\top(P-B) \geq 0$$

$$-\theta(i) \geq -W \quad, \quad \theta(i) \geq 0 \quad \forall i = 1, ..., d.$$

Fix any state action pair $(s', a') \in S \times A$ and change the constraint $\mu(s', a') \geq \delta$ for $\mu(s', a') \geq \delta + \gamma_{s',a'}$. Let $obj(\gamma_{s',a'})$ be the optimal value of the above LP with the constraint is replaced by $\mu(s', a') \geq \delta + \gamma_{s',a'}$. Let $\mu^*(\gamma_{s',a'})$ be the optimal solution to this problem. Let $\bar{\gamma}_{s',a'}$ be the maximum value of $\gamma_{s',a'}$ such that the LP above is feasible.

Some remarks are in order. First, for any $\gamma_{s',a'} \in [0, \bar{\gamma}_{s',a'}]$, it holds that $obj(\gamma_{s',a'}) \geq 0$. Second, $obj(\gamma_{s',a'})$ is a convex and increasing function in $\gamma_{s',a'}$. Third, a subgradient of $obj(\gamma_{s',a'})$ is given by the optimal dual variable associated with the constraint $\mu(s', a') \geq \delta + \gamma_{s',a'}$. Let us call this optimal dual variable $\lambda^*(\gamma_{s',a'})$. Since the above LP's objective is equivalent to $\|\mu - \Phi\tilde{\theta}\|_1$, using triangle inequality we have that $obj(\bar{\gamma}_{s',a'}) \leq \|\mu^*(\bar{\gamma}_{s',a'})\|_1 + \|\Phi\tilde{\theta}\| \leq 1 + \|\Phi\tilde{\theta}\| \leq 2$, where the last inequality holds since $(\Phi\tilde{\theta})^\top 1 = 1$.

We are ready to upper bound $\lambda^*(0)$ which is the subgradient of $obj(\gamma_{s',a'})$ for $\gamma_{s',a'} = 0$. Since $obj(\gamma)$ is an increasing function, we can upper bound $\lambda^*(0)$ with the slope of the line that passes through the points $(0, obj(0))$ and $(\bar{\gamma}_{s',a'}, obj(\bar{\gamma}_{s',a'}))$. The slope of this line is $\frac{obj(\bar{\gamma}_{s',a'}) - obj(0)}{\bar{\gamma}_{s',a'}}$. We have that

$$\lambda^*(0) \leq \frac{obj(\bar{\gamma}_{s',a'}) - obj(0)}{\bar{\gamma}_{s',a'}} \leq \frac{2 - obj(0)}{\bar{\gamma}_{s',a'}} \leq \frac{2}{\bar{\gamma}_{s',a'}},$$

where the last inequality holds since $\|\cdot\|_1 \geq 0$.

Let us now discuss in more detail the quantity $\bar{\gamma}_{s',a'}$. It turns out to be problem-dependent. For example, consider an MDP such that regardless of the action chosen by the player, it transitions to any state with equal probability and there is only one action at each state, then $\bar{\gamma}_{s',a'} = \frac{1}{|S|}$. Thus, the bound for $c_{S,A}$ becomes $c_{S,A} \leq 2|S|$, which depends linearly on $|S|$. Consider another example: suppose the MDP is such that for any state, there exists an action that allows us to remain in that state with probability 1 (a concrete case is the Markovian multi-armed bandit problem with the "retirement" option, see Whittle [41], Weber [40]). This implies that we can make the occupancy measure equal to a vector consisting of zeros of dimension $|S||A|$ with a 1 on any desired entry. Then, the analysis above shows that $\bar{\gamma}_{s',a'} = 1$.