[Reviews · NeurIPS 2019]

Reviewer 1



-- The rebuttal answered all my theoretical questions. I still feel that the work would be greatly improved by adding numerical experiments. -- This paper considers regret minimization in Markov decision processes (MDP). In particular, the authors refer to a specific setting called 'online MDP', where the dynamics, that is, the transition probabilities, are known while the reward is not. Regret minimization then refers to the idea to minimize the regret'' given that rewards could be chosen/observed in an adversarial manner. The authors start with a (rather technical) introduction, pose related work, and explain the main ideas based on concise preliminaries. Afterwards, an extension to large state spaces by using approximate occupancy measures and thereby avoiding concrete state-mappings is provided. In general, I found the paper very well written and good to follow. The topic seems relevant - however, I would have liked a more thorough motivation for the setting where the transitions probabilities are already known while the reward is not. The authors name a few applications (on page 2), but no further explanation is given. That leads to the second problem: No experimental evaluation is provided. While I believe that the linear programming approach will work nicely, it would be good to see how the work performs in practice. That is even more true for the approximate approach for large state spaces. Finally, the authors are not aware of some related work, see for instance Chatterjee, Krishnendu. "Markov decision processes with multiple long-run average objectives." International Conference on Foundations of Software Technology and Theoretical Computer Science. Springer, Berlin, Heidelberg, 2007. Křetínský, Jan, Guillermo A. Pérez, and Jean-François Raskin. "Learning-based mean-payoff optimization in an unknown MDP under omega-regular constraints." arXiv preprint arXiv:1804.08924 (2018). A similar setting has also been considered in Junges, Sebastian, et al. "Safety-constrained reinforcement learning for MDPs." International Conference on Tools and Algorithms for the Construction and Analysis of Systems. Springer, Berlin, Heidelberg, 2016. I also have some technical questions: 1. Is an upper bound on the reward required? In that case, also approaches (expliting dual MDP formulations) from robust/uncertain MDPs could be interesting to compare with, see for instance: Wiesemann, Wolfram, Daniel Kuhn, and Berç Rustem. "Robust Markov decision processes." Mathematics of Operations Research 38.1 (2013): 153-183. 2. Algorithm 1 basically just performs online updates based on the occupancy measure so far, that is on the history of the current policy. Is there not more information that could be exploited? 3. As far as I see a discount factor is not incorporated. Can the approaches be easily adapted?

Reviewer 2



This paper studies algorithms for average-cost MDPs with discrete states and actions, known transition dynamics, bounded adversarial rewards, and exponentially fast mixing. The problem setting and initial regret bound were described in the work of [15]. Their work and follow-up papers use an experts algorithm in each state, with losses corresponding to Q-functions. The algorithm proposed in this paper is different in that works in the dual: it optimizes the state-action occupancy measure subject to dynamics constraints, using FTRL. Compared to [15], the obtained regret bound has an improved dependence on mixing time, but it includes an extra ln|S| term. For MDPs with a large state-space, the authors approximate the occupancy measure using a linear function. This requires several changes: relaxing constraints and adding them as a penalty, and generating stochastic subgradient in order to reduce computational complexity to be polynomial in the number of features rather than |S| and |A|. The corresponding regret bound is with respect to a restricted class of policies, for which the occupancy measure can be expressed as a linear function of the same features. Originality: The proposed algorithm and analysis follow multiple ideas from previous work. In the tabular case, the approach is quite similar to [13]. The authors say that there are some gaps in the proof of Lemma 1 in [13]. I looked at the paper [13] and also contacted the authors, and it seems like this problem is in fact just a typo. Thus I think a clearer statement of contribution in the context of the work of [13] is needed. The computationally efficient version with large MDPs and linear function approximation is more novel. Quality: The paper seems technically correct; I did not verify all of the proofs. Clarity: The paper is well-written and well-organized. However, the authors should clarify the difference between this work and [13], in terms of both the algorithm and analysis. It would also be useful to clarify the benefits, if any, of working in the dual (i.e. optimizing the state-occupancy measure). In the tabular MDP, it adds a dependence on ln|S|. In the large MDP, presumably one could similarly use a linear Q-function, and not have to deal with dynamics constraints. As an aside, in the LQR problem with adversarial costs and known dynamics (https://arxiv.org/abs/1806.07104), there exist some advantages of working in the dual SDP: in the primal, feasible suboptimal solutions do not necessarily correspond to stabilizing controllers. After reading the rebuttal: My overall opinion has not changed much. Yes, the error in [13] is about |S| vs |S||A|. Given that the motivation for the paper is "a setting often encountered in practice, where the state space of the MDP is too large to allow for exact solutions", I agree with the other reviewers that it would be useful to at least provide an example of such a problem, as well as some numerical simulations. Some of the practical examples the authors mention in the rebuttal and list in the introduction *do not* correspond to the studied setting. For example, in queueing networks, the cost function is not unknown/adversarial, but in fact known, and a deterministic function of the state (it's the number of jobs in the system).

Reviewer 3



Summary: This work is a continuation of a line of works on online Markov decision processes (Online MDPs). The main focus of this work is to tackle Online MDPs in very large state spaces. It considers the full-information setting where the dynamics of the MDP is known, while the reward function for each time-step may change adversarially, but is revealed to the learner after each decision. The underlying MDP dynamic has a fast mixing property. The goal of the learner is to maximize the total reward of horizon T. The authors design two algorithms, Alg. MDP-RFTL and Alg. Large-MDP-RFTL. The paper is written clearly. The quality of the analysis is above average in my opinion. On the other hand, the topic of Online MDPs has been considered by several previous works, eg., [15] (Even-Dar et al. 2009), [28] (Neu et al. 2014), [13] (Dick et al. 2014), etc. In terms of the scope, while several of the works above also considered the bandit setting, this paper only deals with the full-information setting. The analysis and the algorithms follow several techniques from [15] (Even-Dar et al. 2009), [3] (Abbasi-Yadkori), etc. The first algorithm Alg. MDP-RFTL has regret bound of order \tilde{O}(\sqrt{\tau(ln|S|+ln|A|)T}ln(T)). The idea is to use Regularized-Follow-The-Leader algorithm on the feasible set stationary distributions of the MDP, as in [13] (Dick et al. 2014). The regret bound for Alg. MDP-RFTL is \tilde{O}(\sqrt{\tau(ln|S|+ln|A|)T}ln(T)), which has an extra dependence on |S|, but in the same time has a better dependence on \tau than the SOTA regret bound in [15]. This algorithm requires the knowledge of the mixing time \tau. The authors design the approximate algorithm, Alg. Large-MDP-RFTL, applicable for large state spaces with computational complexity independent of the state space size, with regret \tilde{O}(\sqrt{T}) compared against the best hindsight in a specified subspace of policies/stationary distributions. Alg. Large-MDP-FRTL also requires the knowledge of the mixing time \tau. The approximate algorithm uses projected stochastic gradient ascent (PSGA) sub-routine to avoid the computational complexity to be dependent on the size of the state and action space. However the number of iterations K(t) needed for the sub-routine PSGA for each time-step t has to be large enough. K(t) is of the order \tilde{O}(t^4\tau^8 T^3) (for K(t) defined in line 621), which may be large, when t, \tau, T are large. Could the authors give some discussion on this? Thanks. (Also the K(t) specified in line 254 in Theorem 2 is different from that in the proof in line 621.) Another issue is about the unknown constant c, which bounds the infinity-norm of the optimal dual variable \lambda^* for the dual problem of Primal 1. The c’s dependence on |S| and |A| is not very clear, which may affect the regret bound \tilde{O}(c\sqrt{T}). (Only a few special cases were discussed in Appendix B, including an example with c\leq 2|S|.) Perhaps more concrete examples may be needed to better assess the quality of the regret bound in terms of c. Considering the real-world applicability of the Alg. Large-MDP-RFTL: 1) First, the theoretical mixing time assumption of the MDPs may not be easily imposed on real-world problems. (While we also assume the dynamics of the MDP is given.) This mixing time \tau in addition appears in the Large-MDP-RFTL hyper-parameters H(t) and K(t). Thus, in practice, one must be able to estimate the mixing time accurately. Hence, if merely estimated approximately, a discussion should be preferable to see how this might affect the algorithm. (More real-world examples in this online MDP setting may help to enhance the paper's impact.) 2) Second, the choice of \Phi should have an impact on the algorithm Large-MDP-RFTL, since the comparator in hindsight is restricted on the subspace determined by \Phi. (Small regret may not eliminate possible poor performance when compared against ill-picked policy linear subspace.) In retrospect, it may be nice to try to include some discussion on choosing the best linear subspace. 3) Third, the constant c in the regret bound \tilde{O}(c\sqrt{T}) for Large-MDP-RFTL is not clear, at the moment, in its dependence on the size of state and action spaces. Based on these concerns, perhaps, it is preferable to supplement the theoretical results with some experiments for this online MDP setting. About the constraint set for \theta, there is inconsistency for the definition of the constant W, shown below: 1) (W as 1-norm bound for \theta): The constraint set \Theta for \theta is defined in line 204-205, where the constant W>0 bounds the 1-norm of \theta. (||\theta||_1\leq W). Then W can also serve as 2-norm bound for \theta. In line 555, ||\theta||_2 \leq W is used, for applying Theorem 3. 2) (W defined as such that 0\leq \theta(i) \leq W \forall i\in [d]): In the proof of Lemma 11, by recasting the 1-norm minimization as a LP problem Primal 1, the condition \theta \in \Theta is rewritten as -\theta(i) \geq –W \forall i \in [d]; \theta(i) \geq 0 \forall i \in [d]. Also in line 571 and 578, it says that “since all entries of \Phi and \theta are nonnegative…” In the proof of Lemma 1, W is also used in the same way. There is an error in Lemma 16: In the proof for the upper bounds for R^\delta, , under line 580, the proof has a mistake: although ln(x) \leq x, we are bounding |ln(x)|=-ln(x). Perhaps we may argue that since R^\delta \leq R (as stated in line 579), where R is the negative entropy, both R^\delta and R are upper bounded by 0. In the proof of Lemma 13 for bounding the Lipschitz constant G_F: |ln(dW)| is replaced by dW in the end. However, |ln(dW)| is not necessarily less than dW, unless dW \geq 1 (although we can usually assume it, for large d and W). Suppose dW \geq 1. Keeping the log may be beneficial: ln(dW) is smaller, which may help in the main theorem to reduce dependencies on d and W. Experiments may further help evaluate the algorithms as in [3].

[Author Response · NeurIPS 2019]

# Response to the paper "Large Scale Markov Decision Processes with Changing Rewards"

We thank the reviewers for their careful review and constructive comments. Due to a lack of space we will discuss the major issues below. We will fix all minor issues and typos in the paper as suggested.

**Reviewer #1**: (*Reward Bound*) Yes, an upper bound on the rewards is necessary, which is standard in online convex optimization problems. If the rewards were unbounded, the adversary could select an arbitrarily large reward for an action that is not taken by the decision maker, leading to arbitrarily bad regret bound. (*Discount Factor*) A discount factor can easily be incorporated. Note that we consider a finite horizon problem of $T$ rounds. If the decision maker needed to incorporate a discount factor $\gamma$, it could just scale the rewards for round $t$ by a factor $\gamma^t$. (*Description of Applications and Motivation*) Due to page limit, we had to shorten out descriptions of important applications that fit our framework (line 41-43 in the paper). We note that all of the applications mentioned in our paper are proposed and discussed in the previous literature. Detailed description or references to these applications will be given in the revised version. (*Numerical Evaluation*) We can add numerical evaluation if space permits. (*Reference*) We appreciate the references and will make sure to add them into the paper.

**Reviewer #2**: (*Comparison with [13] (Dick et al. 2014)*): Our approach and that of [13] are similar on a high level as we both cast Online MDPs as Online Linear Optimization problems. However, there are main differences in how the two papers tackle large state-action spaces. The paper [13] aims to efficiently solve subproblems in each time period by using approximate projections onto subsets of the simplex. This approach requires solving quadratic problems with $O(|S||A|)$ variables and constraints. In [13], the authors suggested (paragraph before corollary 4) solving them using interior point methods, but it is well known that interior point methods not scale well in practice for extremely large scale problems. In contrast, our algorithm tries to leverage stochastic subgradients, which are much cheaper to compute. We will include a more detailed comparison with [13] in the revised paper to clarify our contribution. We also thank the review for clarifying the proof issue in [13] with the author. To be specific, we could not understand the proof of Lemma 1 in [13]: in equation (8), since $l_t$ is of dimension $|S||A|$, vectors $\mu^\pi$ and $\nu_t$ should also be of the same dimension. However, later in the same proof, they write $\|\nu_t - \mu^{\pi_t}\|_1 \leq \|\nu_t - \mu^{\pi_{t-1}}\|_1 + \|\mu^{\pi_t} - \mu^{\pi_t}\|_1 = \|(\nu_{t-1} - \mu^{\pi_{t-1}})P^{\pi_{t-1}}\|_1 + \|\mu^{\pi_t} - \mu^{\pi_t}\|_1$. In section 2.3, they specify $P^\pi \in \mathbb{R}^{|S| \times |S|}$, so $\nu_t$ must also have dimension $|S|$, which contradicts the earlier definition that $\nu_t$ was of dimension $|S||A|$. (*Motivation of problem setting.*) We believe a more general problem involves large state-action spaces, time-varying rewards, bandit feedback, and time-varying and unknown transition dynamics. This general problem is obviously very challenging. The focus on this paper was to address the first two points rigorously. In addition, in the introduction (line 41-43) we provide several applications where the rewards are unknown but the system dynamics are known, which fits to the framework of our model. Due to the space limit, we did not expand on these applications, but we will add details and clarifications in the revised paper. (*Motivation for working with occupancy measures.*) We will include a more detailed discussion to motivate our methodology. In particular, we work with occupancy measures as opposed to the $Q$ function or value functions due to the nice properties of the linear program (Eq. 2) in section 3.1. First, the feasible set is a polytope which is a subset of the probability simplex. The diameter of the $n$-dimensional simplex (measured with the negative entropy) is $\ln(n)$, which allows us to get regret bounds that depend on $\ln(|S||A|)$. Second, when the objective in that linear program is time-varying, it is intuitive to resort to existing machinery from Online Convex Optimization, in the expense that some extra work has to be done to bound the MDP-Regret.

**Reviewer #3**: We appreciate your thorough review. (*Definition of $K(t)$*) You are correct that $K(t)$, the number of PSGA iterations, is not negligible. Fortunately, each iteration is relatively cheap as they do not depend on $|S|, |A|$. We would like to point out that, the specified $K(t)$ bound as a polynomial function of $t$ and $T$ is an upper bound on the number of PSGA iterations. By noticing that from round $t$ to round $t+1$ the solution to (7) may not change much (only one term is added to the summation), one may be able to derive a tighter bound. We did not pursue this as we thought it would obscure the main ideas of the proof of Theorem 2. (*Definition of constant $W$*) Thank you for pointing out the inconsistency. We can fix the issue as follows. The constraint set $\Theta$ will be defined as $\{\theta \in \mathbb{R}^d_+ : \|\theta\|_\infty \leq W\}$ (notice the + in $\mathbb{R}^d_+$). Since $\|\theta\|_2 \leq \sqrt{d}\|\theta\|_\infty$, it holds that $\|\theta\|_2 \leq \sqrt{d}W$ on line 555, and we can still apply Theorem 3, adding a factor of $\sqrt{d}$ in the bound of Lemma 15. Notice that in the proof of Lemma 11, the recasting $\theta \in \Theta$ as $-\theta(i) \geq -W \; \forall i = 1, ..., d, \theta(i) \geq 0 \; \forall i = 1, ..., d$ is now consistent. We are also being consistent with lines 571 and 578. (*Proof of Lemma 16*) Thank you for pointing out the error in Lemma 16. It can indeed be fixed as you suggested: since $\theta \geq 0$, $\Phi$ has probability distributions as it columns, and $1^\top \Phi\theta = 1$ defines the set $\Theta^\Phi$, we indeed have $R(\Phi\theta) \leq 0$. By construction $R^\delta(\Phi\theta) \leq R(\Phi\theta) \forall \theta \in \Theta^\Phi$, it holds that $R^\delta(\Phi\theta) \leq 0$. Therefore, the term $\frac{1}{\eta}(1 + \ln(|S||A|))$ in Lemma 16 can be replaced with the tighter bound $\frac{1}{\eta}(\ln(|S||A|))$. (*Proof of Lemma 13*) Thank you, we agree with you on the proof of Lemma 13. We can stop the bounding terms earlier so that we use $\|\nabla_\mu F_t(\mu)\|_\infty \leq t + \frac{t}{\eta}\max\{|1 + \ln(\delta)|, |1 + \ln(Wd)|\}$. This will, as you point out, improve the dependence with respect to $d, W$ in Theorem 2. (*Typos and minor issues*) Thank you very much for the comments and we will include all of them in the revised paper.

[Meta-Review · NeurIPS 2019]

The paper contributes new algorithmic ideas and theoretical results for regret minimization in Markov Decision Processes with known transition kernels but arbitrary cost functions. The reviewers broadly agree that the theoretical and algorithmic techniques introduced by the paper -- using the FTRL online learning idea and the extension to large MDPs via linear function approximation -- are novel, and thus the paper deserves to be published; however, the known-MDP-unknown-cost setting may be somewhat narrow in its applicability in practice.